# Learning Language-grounded Concepts for Self-explainable Graph Neural Networks

## Abstract

We introduce Graph Concept Bottleneck (**GCB**) as a new paradigm for self-explainable Graph Neural Networks. **GCB** maps graphs into a concept space—a *concept bottleneck*—where each concept is a natural language phrase, and predictions are made based on these concepts. Unlike existing interpretable GNNs that primarily rely on subgraphs as explanations, the concept bottleneck provides a more human-understandable form of interpretation. To refine the concept space, we apply the information bottleneck principle to encourage the model to focus on causal concepts instead of spurious ones. This not only yields more compact and faithful explanations but also explicitly guides the model to *think* toward the correct decision. We empirically show that **GCB** achieves intrinsic interpretability with accuracy on par with black-box GNNs. Moreover, it delivers better performance under distribution shifts and data perturbations, demonstrating improved robustness and generalizability as a natural byproduct of concept-based reasoning.

## 1 Introduction

As Graph Neural Networks (GNNs) Kipf & Welling (2017); Veličković et al. (2018); Yun et al. (2019); Xu et al. (2019) demonstrate strong performance in a wide range of real-world applications, including high-stakes domains Wu et al. (2021)—trustworthiness has emerged as a critical concern. One of the most effective ways to enhance trust is to provide transparent interpretations of the prediction process Kakkad et al. (2023). In this context, intrinsic interpretability, which enables models to explain their predictions directly without relying on post-hoc explanations, becomes a particularly desirable property for GNN-based models Miao et al. (2022). Most self-explanable GNNs (SE-GNNs) Miao et al. (2023); Yu et al. (2022); Wu et al. (2022); Dai & Wang (2021); Azzolin et al. (2025); Dai & Wang (2025); Liu et al. (2025); Peng et al. (2024) focus on extracting the most informative yet compressed causal subgraphs, which is assumed to be responsible for the prediction and is used for both decision-making and explanation. However, while such subgraphs are typically smaller and contain less redundant information, they are still graphs—often complex and difficult to interpret. It remains challenging for humans to understand these explanations, especially in domains where expert knowledge is lacking or the graph structure is intricate.

We aim to narrow the gap between model predictions and human understanding by introducing an intermediate representation that is more interpretable than subgraphs. To this end, we propose inserting a *concept bottleneck* layer into the neural network. Specifically, the input graph is mapped to a concept layer that captures its activations over a set of semantically meaningful concepts. These concept activations are then mapped to the label space through a few feedforward layers for label prediction. In this way, the concept activations serve a dual purpose: they drive the prediction and simultaneously provide explanations for it. Although the general workflow is straightforward, adapting it to graph prediction tasks is non-trivial and introduces several challenges: (1) *Concept selection:* It is labor-intensive to predefine concept sets relevant to the prediction task, and graphs often represent abstract structures (e.g., social networks), making it difficult to define and select meaningful, human-interpretable concepts. (2) *Concept alignment:* It remains unclear how to effectively map the input domain (graphs) to the concept domain (language). Unlike in vision-language tasks—where models like CLIP Radford et al. (2021) provide off-the-shelf alignment—graphs exhibit irregular structures and high variability across domains, and no such readily applicable model exists. Consequently, a concept predictor that minimizes information leakage Havasi et al. (2022); Sun et al. (2024) must be carefully designed to ensure faithful explanations.

In light of these challenges, we propose **Graph Concept Learning** (**GCB**) as a new paradigm for interpretable graph learning. **GCB** consists of three modules. First, we pre-train a universal graph encoder using self-supervised *Contrastive Concept–Graph Pretraining*, which aligns graph representations to the concept space. The resulting encoder can be applied across different downstream datasets. Next, we construct an initial concept space through *LLM-enhanced Concept Retrieval*, eliminating the need for manual annotation. This space is further refined by filtering out spurious concepts while retaining causal ones via *Information-Constrained Bottleneck Optimization*. Finally, we train a predictor that operates over the refined concept space to make task-specific predictions.

We conduct extensive experiments to evaluate the effectiveness of **GCB**. Supported by strong empirical evidence, we highlight two major contributions of **GCB**: (1) *A language-based interpretable graph learning framework.* To the best of our knowledge, **GCB** is the first self-explainable graph learning framework that projects graph inputs into a language space during prediction. This demonstrates the potential of actively incorporating natural language as an integral part of the reasoning process, enabling more interpretable and transparent deep learning on graph-structured data. Importantly, the explanations provided are faithful to the model's decision process and accurately reflect the semantic meaning of the language concepts, without information leakage from labels to the concept space. (2) *A robust baseline for node-level classification.* We show that **GCB** performs competitively with SOTA GNNs in standard settings, and its advantages become more pronounced under distribution shifts and data perturbations, establishing a strong baseline for robust and generalizable graph learning.

## 2 RELATED WORK

In recent years, there has been growing interest in self-explainable GNNs Miao et al. (2022; 2023); Yu et al. (2021; 2022); Wu et al. (2020; 2022); Dai & Wang (2021); Feng et al. (2022); Azzolin et al. (2025); Dai & Wang (2025); Liu et al. (2025); Peng et al. (2024), where the explainability component is integrated into the prediction process. These methods typically generate informative subgraphs that serve both as explanations and as the basis for predictions. One line of work Miao et al. (2022; 2023); Yu et al. (2021; 2022); Wu et al. (2020) leverages the information bottleneck principle, aiming to extract the most informative yet compact subgraph by optimizing a graph information bottleneck objective. Other approaches Wu et al. (2022); Dai & Wang (2021); Feng et al. (2022) introduce structural constraints to promote interpretability. For example, DIR Wu et al. (2022) decomposes the input graph into causal and non-causal components, enforcing that predictions depend only on the causal part. Despite these advances, most existing methods still rely on subgraphs as explanations, whose interpretability is not always guaranteed. More recently, researchers have begun exploring alternative forms of explanation. For instance, Bechler-Speicher et al. (2024) proposes Graph Neural Additive Networks, where the relationships between the input graph and the target variable can be directly visualized. Sengupta & Rekik (2025) encodes interpretable cues (e.g., degrees, centrality) into a context vector, which is then mapped into an explanation vector. Müller et al. (2023) employs decision trees to build rule-based predictors that are understandable to humans. However, none of these works employ natural language as a medium for explanations.

## 3 GRAPH CONCEPT BOTTLENECK

### 3.1 CONTRASTIVE CONCEPT–GRAPH PRETRAINING

We propose Contrastive Concept–Graph Pretraining (CCGP), which pretrains a multimodal model to align graph and text representations in a shared space. CCGP is specifically designed to enhance graph-to-concept alignment and can be applied universally across diverse datasets and domains.

**Pretraining data.** We collect unlabeled graph data from diverse domains to construct the pretraining dataset for CCGP. Prior work Wang et al. (2024); Chen et al. (2024); Tang et al. (2024) has demonstrated the remarkable ability of LLMs to understand and reason over graph-structured data. Motivated by this, we leverage LLMs to generate self-supervised concept annotations. For each dataset, we sample $m$ instances; for each instance $x_i$, we query GPT-3.5 Brown et al. (2020) to generate a list of associated concepts (see Appendix B.1 for prompt details). We collect all instance–concept list pairs $\{(x_i, \mathcal{C}_i)\}$ for future procedures.

We augment the pretrained data to improve the robustness of the pretrained model against noise and structural perturbations. For each instance $x_i$ we create a set of perturbed graphs $\mathcal{X}_i^{\text{aug}} = \left\{ \tilde{x}_i^{(1)}, \tilde{x}_i^{(2)}, \ldots, \tilde{x}_i^{(M)} \right\}$, where each $\tilde{x}_i^{(m)}$ is constructed by perturbing the $k$-hop neighborhood of $v_i$. Each augmented view is obtained by randomly dropping/adding edges with ratios $\rho_{\text{add}}$ and $\rho_{\text{drop}}$ from the original graph, and the augmented instance–concept list pairs are obtained as $\{(\mathcal{X}_i^{\text{aug}}, \mathcal{C}_i)\}$

**Encoders.** The pretrained model consists of a graph encoder and a text encoder. The graph encoder $f_\theta^{\text{GNN}}(\cdot)$ with trainable parameter $\theta$ is responsible for capturing both the feature attributes and topological structure of the graph, and it should generalize well to downstream graph data, potentially from different datasets. More expressive architectures, such as Graph Transformers, are capable of modeling rich semantics and complex patterns, but they are more prone to overfitting than smaller GNNs like GCN. We adopt a pretrained Sentence-BERT Reimers & Gurevych (2019) model as the text encoder $f^{\text{LM}}(\cdot)$, with parameters kept frozen throughout training. This choice leverages the model's strong general semantic capabilities, while avoiding the computational cost and overfitting risks associated with fine-tuning large language models on limited data.

**Set2set contrastive learning.** For each graph (node) instance $x_i$, we have a set of augmented views $\mathcal{X}_i^{\text{aug}} = \left\{ \tilde{x}_i^{(1)}, \tilde{x}_i^{(2)}, \ldots, \tilde{x}_i^{(M)} \right\}$ and a set of concepts $\mathcal{C}_i = \{c_{i1}, c_{i2}, \ldots, c_{iK}\}$. This results in a *set-to-set* alignment problem, where each augmented graph view $\tilde{x}_i^{(m)}$ is semantically aligned with every concept $c_{ij}$ in $\mathcal{C}_i$. During training, we construct positive pairs by sampling from the Cartesian product of the augmented views and the concept set. Specifically, for each instance $x_i$, we sample a subset of positive pairs: $P_i = \left\{ \left( \tilde{x}_i^{(m)}, c_{ij} \right) \mid m \in \mathcal{M}_i, \ j \in \mathcal{K}_i \right\}$ where $\mathcal{M}_i \subseteq \{1, 2, \ldots, M\}$ and $\mathcal{K}_i \subseteq \{1, 2, \ldots, K\}$ are sampled subsets of augmented views and concepts, respectively. For each pair $(\tilde{x}_i^{(m)}, c_{ij}) \in P_i$, we compute embeddings the graph embedding $z_i^{(m)} = f_\theta^{\text{GNN}}(\tilde{x}_i^{(m)})$ and the text embedding $z_{ij}^{\text{concept}} = f^{\text{LM}}(c_{ij})$. We then apply a contrastive loss based on the InfoNCE van den Oord et al. (2018) formulation to maximize the similarity between positive pairs while minimizing similarity to negative pairs in the batch. The contrastive loss for each positive pair is defined as:

$$\mathcal{L}_{i,j}^{(m)} = -\log \frac{\exp\left( \text{sim}\left( z_i^{(m)}, z_{ij}^{\text{concept}} \right) / \tau \right)}{\sum\limits_{(k,l,n) \in \mathcal{B}} \exp\left( \text{sim}\left( z_i^{(m)}, z_{kl}^{\text{concept}(n)} \right) / \tau \right)}, \tag{1}$$

where $\text{sim}(\cdot, \cdot)$ denotes cosine similarity, $\tau$ is the temperature parameter, and $\mathcal{B}$ is the set of all (view, concept) pairs in the current batch. Overall, we formulate the learning objective as minimizing the following contrastive loss with respect to model parameters $\theta$:

$$\theta^* = \arg\min_\theta \ \mathcal{L}(\theta) = \frac{1}{\sum_i |\mathcal{M}_i||\mathcal{K}_i|} \sum_i \sum_{m \in \mathcal{M}_i} \sum_{j \in \mathcal{K}_i} \mathcal{L}_{i,j}^{(m)}(\theta). \tag{2}$$

This set-to-set sampling and contrastive learning ensure that the model learns to robustly align multiple augmented views of each graph with multiple semantically meaningful concepts, improving its generalization across diverse graph data. We donate the optimized graph encoder as $f^{\text{GNN}}(\cdot)$. The parameters of $f^{\text{GNN}}(\cdot)$ are frozen during the subsequent training process to ensure independence from label supervision, thereby minimizing potential information leakage and preserving the faithfulness of the explanations.

## 3.2 LLM-EMPOWERED CONCEPT RETRIEVAL

Given the strong ability of LLMs in domain knowledge Lee et al. (2024), abstraction & pattern recognition Lee et al. (2025b), and contextual reasoning Zhang et al. (2024), we construct the concept space through two complementary approaches:

**Global Concept Proposal.** We expect LLMs to identify concepts to distinguish between classes when instructed appropriately. Specifically, we provide detailed description of the dataset and ask the LLM to generated an initial set of revelent concepts for each class. See B.1 for prompt details.

**Instance-Based Concept Extraction.** While Global Concept Proposal offers broader semantic coverage and reflects domain-level priors, it may overlook dataset-specific nuances or generate

abstract concepts that lack clear anchoring in the input graphs. Observing the richness of the training data and LLMs' ability to perform contextual reasoning and summarize fine-grained patterns on graph data Wang et al. (2024); Chen et al. (2024); Tang et al. (2024), we propose to ask LLMs to recognize relevant concepts given sampled graph instances (see Appendix B.1 for prompt details). Specifically, we sample $m$ graph instances from each class and apply the prompt to each sampled graph instance, resulting in a large set of candidate concepts. We then identify a subset of concepts that are highly relevant to each class, distinct from those used by other classes, and useful for improving class discrimination. Please refer to the Appendix B.2 for the details of this process.

To control the quality and size of the concept set we perform several filtering steps to remove redundant or irrelevant concepts. Details of the filtering process are provided in B.2. We denote the filtered concepts from the Global Concept Proposal and Instance-Based Concept Extraction $\mathcal{C}^{\text{glob}}$ and $\mathcal{C}^{\text{glob}}$, separately. We combine $\mathcal{C}^{\text{glob}}$ and $\mathcal{C}^{\text{inst}}$ as the candidate concept set as $\mathcal{C}^{\text{candidate}}$.

### 3.3 Information-Constrained Concept Optimization

The retrieved concept space in Section 3.2 may contain too many concepts, and some of them could be irrelevant or spurious, hindering both the explanablity and the generalizability of the model. To address this, we adopt the Information Bottleneck (IB) Alemi et al. (2017) criteria to encourage the model to rely on a sparse set of concepts that are causal to the prediction.

**Definition 1** *The Information Bottleneck criteria is generally formulated as $I(Z;Y) - \beta I(Z;X)$, which seeks a representation $Z$ that is both informative and compressed: maximizing mutual information with the label $Y$ while minimizing mutual information with the input $X$. A larger $\beta$ results in stronger compression, encouraging $Z$ to retain only the most essential information for predicting $Y$.*

In our model, we apply the IB objective to learn a gate vector $g$ over the fixed concept space. Specifically, for each concept $j$, we learn a soft gate:

$$g_j = \sigma\big(\text{MLP}^{\text{gate}}_\phi(f^{\text{LM}}(c_j))\big), \tag{3}$$

where $\text{MLP}^{\text{gate}}_\phi(\cdot)$ is a learnable multi-layer perceptron applied to the concept embedding $f^{\text{LM}}(c_j)$, and $\sigma(\cdot)$ denotes the sigmoid activation function. We then apply the gate vector to the concept activation vector of each instance $i$ as $z_i = g \odot c_i$, where $\odot$ denotes element-wise multiplication, $c_i$ is the concept activation vector for instance $i$, and $z_i$ is the masked concept vector passed to the classifier. Following the IB principle, we optimize the following objective:

$$\min \frac{1}{N} \sum_{i=1}^{N} \mathbb{E}_{\epsilon \sim p(\epsilon)} \left[ -\log q(y_i \mid z_i) \right] + \beta \, \text{KL}\left( p(z_i \mid x_i) \, \| \, r(z_i) \right), \tag{4}$$

where the first term promotes predictive accuracy, and the second term minimizes the Kullback–Leibler (KL) divergence between the concept representation $z_i$ and the input $x_i$, effectively penalizing their mutual information and encouraging a more compressed and focused representation. In our framework, the prediction function $q(y_i \mid z_i)$ is parameterized by a trainable multi-layer perceptron $\text{MLP}^{\text{cls}}_\psi$, which takes the masked concept vector $z_i$ as input. The gate vector $g$, which determines the masking over the concept activations, is computed by a separate network parameterized by $\phi$. Since $z_i$ is deterministically computed and we do not model a distribution over $p(z_i \mid x_i)$, we approximate the KL divergence term with a deterministic sparsity regularizer. In particular, we use an $L_1$ penalty, which encourages the gate values to shrink toward zero, effectively suppressing irrelevant concepts. This results in a sparse, interpretable concept selection, aligning with the Information Bottleneck's objective of compressing the intermediate representation while retaining task-relevant information. Thus, the training objective in Equation 4 becomes:

$$\min_{\phi, \psi} \frac{1}{N} \sum_{i=1}^{N} \mathcal{L}_{\text{CE}}\left( \text{MLP}^{\text{cls}}_\psi(z_i), y_i \right) + \beta \, \|g\|_1, \tag{5}$$

where $\mathcal{L}_{\text{CE}}$ denotes the cross-entropy loss between the predicted label distribution and the ground-truth label. While the gates are continuous and soft during training, for interpretability, we require a discrete selection of concepts. To achieve this, after the IB training phase, we freeze the learned gate vector $g$ and select the top-$K$ concepts with the highest gate values $\mathcal{C}^{\text{selected}} = \text{Top-K}_j(g_j)$, where $\mathcal{C}^{\text{selected}}$ denotes the final set of selected concepts.

### 3.4 PREDICTOR LEARNING

We use the selected concept set $\mathcal{C}^{\text{selected}}$ to make final predictions. For each instance $x_i$, we compute a concept activation vector $\mathbf{x}_i^{\mathcal{C}} \in \mathbb{R}^{|\mathcal{C}^{\text{selected}}|}$, where each element is defined as:

$$\mathbf{x}_i^{\mathcal{C},(j)} = \text{sim}\left(f^{\text{GNN}}(x_i),\ f^{\text{LM}}(c_j)\right),$$

with $c_j \in \mathcal{C}^{\text{selected}}$ denoting the $j$-th selected concept. $\text{sim}(\cdot, \cdot)$ denotes a similarity metric such as cosine similarity. We then train a predictor using only the concept activation vector $\mathbf{x}_i^{\mathcal{C}}$ as input. Specifically, we use a multi-layer perceptron (MLP) classifier $\text{MLP}_{\theta}^{\text{gate}}$, parameterized by $\theta$, to predict the label $y_i$. The optimization objective is to minimize the cross-entropy loss over the training set:

$$\theta^* = \arg\min_{\theta} \ \frac{1}{N} \sum_{i=1}^{N} \mathcal{L}_{\text{CE}}\left(\text{MLP}_{\theta}^{\text{gate}}\left(\mathbf{x}_i^{\mathcal{C}}\right),\ y_i\right),$$

where $\mathcal{L}_{\text{CE}}$ denotes the standard cross-entropy loss, and $N$ is the number of training instances.

## 4 EXPERIMENTS

We investigate the *robustness* and *interpretability* of **GCB**. First, we evaluate its utility across datasets from different domains and under varying conditions to assess robustness and generalizability to distribution shifts and data perturbations (see Section 4.2). We then analyze the sensitivity of **GCB** to concept size and the choice of graph encoders (Section 4.3). Next, we take a closer look at how the relevance of concepts may affect model performance and potentially lead to information leakage (Section 4.4). Finally, we conduct a case study to visualize how **GCB** provides intuitive explanations for its predictions via the concept bottleneck layer (Section 4.5).

### 4.1 DATASETS

Following the practice in GraphCLIP (Zhu et al., 2025), we use non-overlapping datasets from diverse domains to pretrain the Graph-Concept Alignment model. The graph data used for pre-training is required to be of the same type as the downstream datasets to ensure transferability. In this work, we focus on *text-attributed graphs*, where node attributes are textual descriptions of their contents.

**Source datasets.** We employ five source datasets: `Pubmed` (Sen et al., 2008) is citation network in Biomedicine domain, `Ele-Computers`, `Sports-Fitness`, `Books-Children`, and `Books-History` (Yan et al., 2023) are co-purchasing networks in e-commerce. For each dataset, we sample 1,000 nodes and query GPT-3.5 Turbo to generate 10 concepts/keywords that appear in each node's ego network, serving as ground truth for the Graph-Concept Alignment task.

**Target datasets.** We use `Cora` (Sen et al., 2008), `Citeseer` (Sen et al., 2008), `Instagram` (Huang et al., 2024), `Reddit` (Huang et al., 2024), and `WikiCS` (Mernyei & Cangea, 2020) as our target datasets. `Cora` and `Citeseer` are citation networks in the Computer Science domain; `Instagram` and `Reddit` are social networks; and `WikiCS` is a Wikipedia article network. We ensure that all target datasets are from different domains than the source datasets to evaluate model generalizability and prevent any data leakage. We evaluate them under three settings:

- *Regular setting.* We randomly split each dataset into training, validation, and test sets such that all sets follow the same data distribution.
- *OOD setting.* We split the dataset to induce distribution shifts between training and test sets. Following (Han et al., 2025), we divide the data into majority and minority classes. During splitting, instances from the majority class are $\gamma$ times more likely to be included in the training/validation set compared to those from the minority class, where $\gamma$ is the upsampling ratio. A higher $\gamma$ results in a greater distribution shift between training and test sets. We set $\gamma \in \{2, 3, 5, 10\}$.
- *Adversarial setting.* Using the same split as the *regular setting*, we perturb the edges in the training set by randomly dropping and adding edges for each node with a perturbation ratio $\rho$. We set $\rho \in \{0.05, 0.1, 0.2, 0.3, 0.5\}$.

For all datasets and settings, we adopt a default train/validation/test split of 20%/20%/50%. We use an *inductive* setting, where test nodes are entirely unseen during training and vice versa. We report the Macro F1 scores and Balanced Accuracy Score (BACC) to evaluate model performance to account for class imbalance. See C.1 and C.2 for further details on the datasets and experimental settings.

## 4.2 MAIN RESULTS

We evaluate **GCB** on target datasets under three different settings. The *regular setting* assesses whether **GCB** can provide intrinsic interpretability to GNNs with minimal loss in model utility. The *OOD setting* contains distribution shifts, and the *perturbation setting* includes structural or feature perturbations. For each setting, we also evaluate a set of SOTA GNN and MLP models, including MLP, GCN Kipf & Welling (2017), GAT Veličković et al. (2018), GraphSAGE (SAGE) Hamilton et al. (2017), and Graph Transformer (GT) Yun et al. (2019), as baselines for comparison. We also compare with self-explainable GNNs including GIB Yu et al. (2021), VGIB Yu et al. (2022), DIRGNN Wu et al. (2022), and SEGNN Dai & Wang (2021).

Table 1: Node classification performance in *OOD settings* with upsampling ratio $\gamma = 5$. The best-performing interpretable GNN is underlined, and the overall best-performing method is **bolded**.

| Method | Cora F1 (%) | Cora BACC (%) | Citeseer F1 (%) | Citeseer BACC (%) | Instagram F1 (%) | Instagram BACC (%) | Reddit F1 (%) | Reddit BACC (%) | WikiCS F1 (%) | WikiCS BACC (%) |
|---|---|---|---|---|---|---|---|---|---|---|
| MLP | $50.00_{(0.63)}$ | $60.81_{(0.50)}$ | $38.44_{(0.99)}$ | $54.34_{(1.10)}$ | $35.42_{(0.58)}$ | $51.69_{(0.26)}$ | $16.38_{(0.52)}$ | $51.33_{(0.51)}$ | $54.31_{(0.31)}$ | $65.30_{(0.35)}$ |
| GCN | $55.10_{(0.66)}$ | $64.03_{(1.05)}$ | $46.52_{(0.92)}$ | $59.64_{(0.92)}$ | $36.98_{(0.84)}$ | $50.01_{(0.70)}$ | $12.91_{(0.38)}$ | $48.48_{(0.21)}$ | $\mathbf{59.04}_{(1.66)}$ | $\mathbf{68.38}_{(1.73)}$ |
| GAT | $51.30_{(1.27)}$ | $61.52_{(1.15)}$ | $45.62_{(1.01)}$ | $58.77_{(0.87)}$ | $33.31_{(0.59)}$ | $50.18_{(0.41)}$ | $12.93_{(0.30)}$ | $49.34_{(0.18)}$ | $57.05_{(1.00)}$ | $64.53_{(0.85)}$ |
| SAGE | $44.26_{(1.78)}$ | $53.95_{(1.65)}$ | $30.87_{(0.41)}$ | $48.42_{(0.47)}$ | $31.46_{(0.20)}$ | $48.27_{(0.18)}$ | $13.38_{(0.17)}$ | $49.18_{(0.47)}$ | $51.87_{(1.24)}$ | $62.06_{(1.37)}$ |
| GT | $38.26_{(1.64)}$ | $48.66_{(1.36)}$ | $28.38_{(1.58)}$ | $48.22_{(0.77)}$ | $30.90_{(0.43)}$ | $48.06_{(0.37)}$ | $12.64_{(0.54)}$ | $48.62_{(0.26)}$ | $54.06_{(0.84)}$ | $62.50_{(0.77)}$ |
| DIR-GNN | $23.07_{(2.70)}$ | $43.18_{(2.13)}$ | $15.31_{(1.33)}$ | $42.93_{(1.00)}$ | $26.74_{(0.00)}$ | $50.00_{(0.00)}$ | $8.46_{(0.00)}$ | $50.00_{(0.00)}$ | $22.93_{(1.35)}$ | $42.11_{(0.42)}$ |
| GIB | $19.23_{(4.10)}$ | $40.24_{(3.48)}$ | $15.52_{(1.79)}$ | $42.31_{(1.19)}$ | $26.75_{(0.01)}$ | $50.00_{(0.01)}$ | $8.47_{(0.03)}$ | $50.01_{(0.01)}$ | $24.98_{(1.27)}$ | $39.15_{(1.12)}$ |
| VGIB | $44.56_{(6.43)}$ | $57.06_{(4.66)}$ | $22.26_{(6.35)}$ | $47.72_{(3.26)}$ | $26.74_{(0.00)}$ | $50.00_{(0.00)}$ | $8.46_{(0.00)}$ | $50.00_{(0.00)}$ | $56.02_{(1.76)}$ | $64.14_{(1.25)}$ |
| SEGNN | $30.68_{(2.91)}$ | $48.75_{(1.96)}$ | $19.92_{(2.80)}$ | $42.89_{(1.55)}$ | $26.74_{(0.00)}$ | $50.00_{(0.00)}$ | $8.46_{(0.00)}$ | $50.00_{(0.00)}$ | $34.97_{(1.26)}$ | $50.71_{(1.01)}$ |
| **GCB** | $\mathbf{56.63}_{(1.38)}$ | $\mathbf{66.71}_{(0.99)}$ | $\mathbf{60.19}_{(0.61)}$ | $\mathbf{67.12}_{(0.60)}$ | $\mathbf{56.80}_{(0.23)}$ | $\mathbf{58.47}_{(0.38)}$ | $\mathbf{48.16}_{(0.25)}$ | $\mathbf{63.07}_{(0.99)}$ | $56.36_{(0.46)}$ | $\underline{67.57}_{(0.73)}$ |

Table 2: Node classification performance in *perturbation settings* with upsampling ratio $\rho = 0.3$. The best-performing interpretable GNN is underlined, and the overall best-performing method is **bolded**.

| Method | Cora F1 (%) | Cora BACC (%) | Citeseer F1 (%) | Citeseer BACC (%) | Instagram F1 (%) | Instagram BACC (%) | Reddit F1 (%) | Reddit BACC (%) | WikiCS F1 (%) | WikiCS BACC (%) |
|---|---|---|---|---|---|---|---|---|---|---|
| MLP | $43.08_{(0.45)}$ | $56.55_{(0.37)}$ | $37.77_{(0.53)}$ | $53.77_{(0.59)}$ | $36.69_{(0.45)}$ | $52.57_{(0.19)}$ | $16.75_{(0.84)}$ | $51.11_{(0.44)}$ | $53.70_{(0.32)}$ | $64.95_{(0.45)}$ |
| GCN | $56.21_{(1.25)}$ | $66.49_{(0.85)}$ | $46.45_{(1.42)}$ | $58.14_{(1.61)}$ | $43.87_{(3.28)}$ | $54.11_{(0.85)}$ | $16.64_{(0.77)}$ | $50.43_{(0.76)}$ | $61.45_{(0.38)}$ | $65.64_{(0.87)}$ |
| GAT | $52.64_{(1.33)}$ | $61.70_{(0.71)}$ | $46.55_{(0.87)}$ | $60.54_{(0.58)}$ | $37.07_{(1.09)}$ | $52.38_{(0.43)}$ | $16.51_{(0.49)}$ | $51.64_{(0.39)}$ | $61.58_{(0.44)}$ | $69.76_{(0.49)}$ |
| SAGE | $53.15_{(2.06)}$ | $57.70_{(2.21)}$ | $39.37_{(1.30)}$ | $56.60_{(0.90)}$ | $38.45_{(2.14)}$ | $52.79_{(0.90)}$ | $16.51_{(0.49)}$ | $51.64_{(0.39)}$ | $61.58_{(0.44)}$ | $69.76_{(0.49)}$ |
| GT | $46.13_{(2.23)}$ | $55.15_{(1.80)}$ | $33.36_{(1.69)}$ | $52.65_{(0.93)}$ | $35.36_{(0.78)}$ | $51.70_{(0.27)}$ | $17.29_{(0.50)}$ | $51.31_{(0.12)}$ | $57.88_{(0.95)}$ | $62.61_{(1.60)}$ |
| DIR-GNN | $70.78_{(2.43)}$ | $70.20_{(2.49)}$ | $62.03_{(0.86)}$ | $\underline{64.65}_{(0.70)}$ | $55.56_{(1.43)}$ | $56.45_{(0.64)}$ | $54.13_{(1.52)}$ | $\mathbf{56.35}_{(0.57)}$ | $57.07_{(3.45)}$ | $56.34_{(1.97)}$ |
| GIB | $32.94_{(18.33)}$ | $37.41_{(15.43)}$ | $47.23_{(15.64)}$ | $52.91_{(11.58)}$ | $38.55_{(6.36)}$ | $50.74_{(0.95)}$ | $39.96_{(7.82)}$ | $51.54_{(1.85)}$ | $21.62_{(10.37)}$ | $25.78_{(9.43)}$ |
| VGIB | $20.15_{(26.58)}$ | $26.24_{(23.92)}$ | $54.92_{(20.60)}$ | $57.43_{(17.87)}$ | $39.13_{(0.61)}$ | $50.09_{(0.17)}$ | $34.79_{(3.10)}$ | $50.20_{(0.39)}$ | $58.67_{(24.39)}$ | $59.70_{(22.48)}$ |
| SEGNN | $52.58_{(4.71)}$ | $56.78_{(3.34)}$ | $59.76_{(1.11)}$ | $62.69_{(1.06)}$ | $55.15_{(0.64)}$ | $55.40_{(0.43)}$ | $55.44_{(0.85)}$ | $55.77_{(0.71)}$ | $38.08_{(1.10)}$ | $42.11_{(1.15)}$ |
| **GCB** | $\underline{70.98}_{(0.73)}$ | $\mathbf{71.36}_{(1.07)}$ | $\underline{63.44}_{(0.29)}$ | $63.84_{(0.32)}$ | $\mathbf{56.65}_{(0.26)}$ | $\mathbf{56.61}_{(0.27)}$ | $\underline{55.56}_{(0.74)}$ | $55.58_{(0.77)}$ | $\mathbf{66.08}_{(0.53)}$ | $\mathbf{70.43}_{(0.75)}$ |

*(1)* ***GCB can improve the model generalizability in OOD data.*** We evaluate **GCB** under the *OOD setting* across different upsampling ratios. Due to space constraints, we report the test results in Table 1 for the upsampling ratio $\gamma = 5$; the complete results for all upsampling ratios are provided in E. The results show that **GCB** not only significantly outperforms all self-explainable graph learning methods, but also consistently surpasses state-of-the-art GNNs. We attribute this to **GCB**'s reliance on causal concepts for prediction, which makes it less susceptible to distribution shifts. **GCB** is therefore a strong baseline for improving OOD generalizability in graph learning.

*(2)* ***GCB improves model robustness under training data perturbations.*** We evaluate **GCB** under the *Adversarial setting* with different perturbation ratios. We only report the test results in Table 2 for perturbation ratio $\rho = 0.3$; full results for all perturbation ratios are provided in Appendix E. We observe that while most GNNs perform well under clean conditions, their performance degrades significantly when trained on perturbed data, highlighting their vulnerability to evasion attacks. In contrast, **GCB** demonstrates strong robustness against perturbed train data, while maintaining performance comparable to the model trained on clean data. We attribute this robustness to the use of a pretrained graph encoder trained on augmented data from diverse domains.

*(3)* ***GCB incurs minimal cost in model utility on clean in-distribution data.*** We evaluate **GCB** and baseline methods under the *regular setting*, and report the test BACC scores (averaged over 5 trials)

in Table 4. On three out of five datasets, **GCB** achieves the best performance (in at least one metric) among interpretable GNN methods. Moreover, compared to the overall best-performing model, **GCB** delivers competitive results with only small performance gaps, demonstrating its ability to retain high predictive utility while offering interpretability.

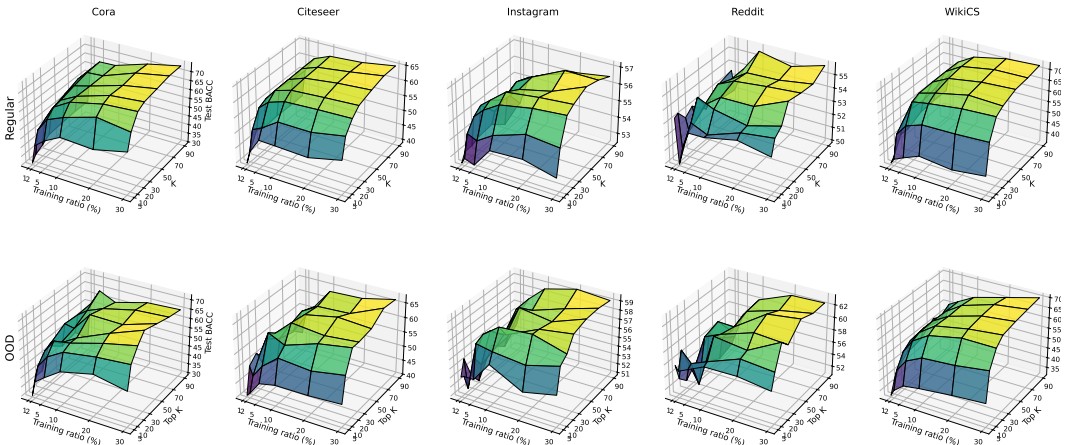

Figure 1: Performance of **GCB** across different concept sizes ($K$) and training ratios (%) on regular splits (top row) and OOD splits (bottom row).

## 4.3 SENSITIVITY ANALYSIS

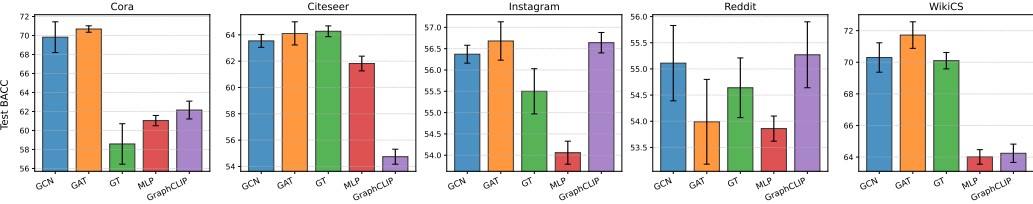

Figure 2: Performance of **GCB** variations using different graph encoders.

**Size of concept set.** The size of the concept set is a critical parameter to consider. Too many concepts can negatively impact the model's interpretability, while too few may reduce its utility by lacking enough information to make accurate predictions. We examine the sensitivity of **GCB** to different concept set sizes $K$, across varying training ratios for each dataset. The results are visualized in Figure 1, where the x-axis represents the training ratio and the y-axis shows the number of concepts. We observe a general trend where the model's performance improves rapidly as the number of concepts increases, but the rate of improvement gradually slows down, eventually plateauing. In the out-of-distribution (OOD) setting, however, increasing the number of concepts may actually hurt performance, particularly at smaller training ratios. Including too many concepts may also hinder the model's generalizability.

**Graph encoders.** We investigate how different graph-text alignment models affect performance. First, we compare various versions of GCB using different graph encoders: GCN (the default), GAT, and Graph Transformer (GT). We also explore the effect of removing the graph structure by replacing the graph encoder with a simple MLP for decoding the concept map. Additionally, we evaluate a pretrained graph foundation model, GraphCLIP, which includes both a graph encoder and a text encoder for graph-text alignment. All results are shown in Figure 2. We observe that GCN consistently performs well across all datasets compared to GAT and GT, suggesting that a simpler architecture may be more stable when pretraining data is limited. The model's performance drops significantly when using the MLP encoder, highlighting the importance of leveraging graph structure for mapping input graphs into the concept space. GraphCLIP performs slightly better on Instagram

and Reddit but considerably worse on the other three datasets. We hypothesize that this is because GraphCLIP aligns graphs to free-form summaries that contain noisy information, which can lead to inaccurate mappings between graphs and their underlying concepts. Moreover, since GraphCLIP jointly trains both the graph and text encoders, the large number of parameters in the text encoder may cause overfitting, especially when the training corpus is small or domain-specific. This limitation could explain why GraphCLIP performs well on Instagram and Reddit—social networks that likely share overlapping topics with its training corpus—but poorly on Cora, Citeseer, and WikiCS, which have little to no topic overlap with the training data.

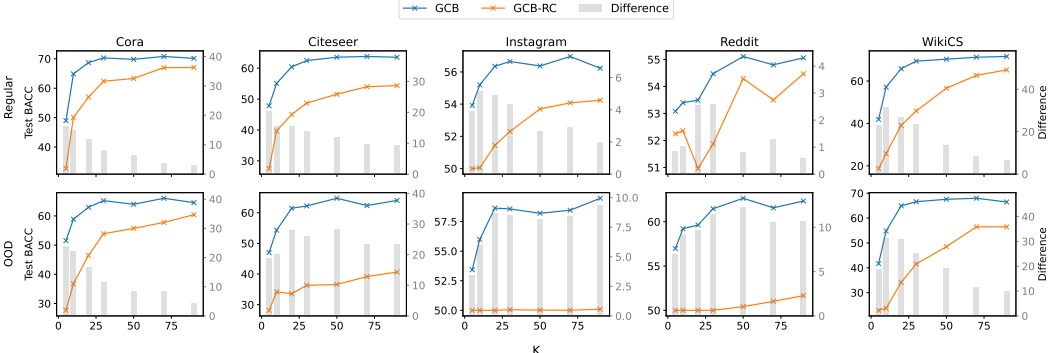

Figure 3: Performance of the original **GCB** compared to its variant with random concepts (GCB-RC) across different concept set sizes, on regular splits (top row) and OOD splits (bottom row).

## 4.4 FAITHFULNESS & INFORMATION LEAKAGE.

While the quality and relevance of the retrieved concept set can be partially validated by the model's decent performance due to its self-explainable nature, there remains a concern about information leakage, which can undermine interpretability and faithfulness Havasi et al. (2022); Sun et al. (2024). When the concept learning model is trained, the label predictor might exploit spurious signals from the concept activations produced by the concept predictor rather than relying on the true semantics of the concepts. In other words, even if the concepts are meaningless or unrelated, the model could still achieve high accuracy by assigning higher activation scores to arbitrary concepts that correlate with the label, providing no true explainable value. Inspired by Mahinpei et al. (2021), although there is no straightforward way to directly measure information leakage, we can evaluate it indirectly by replacing the concepts with random ones. Intuitively, if the model maintains strong performance with random concepts, it suggests the presence of information leakage. We report the performance of **GCB** using both retrieved concepts ("GCB") and random concepts ("GCB-RC") across different numbers of selected concepts $K$, under *regular* and *OOD* settings, shown in Figure 3. The difference between the two is plotted as a gray bar. For the regular split, we observe a general pattern: the performance gap gradually decreases as the concept size increases. Specifically, GCB-RC performs significantly worse with smaller concept sizes but gradually approaches GCB's performance as the concept size grows. This suggests that when the concept set is large enough, the model may rely more on spurious correlations between concept activation patterns and labels. Conversely, when the concept set is small, the spurious patterns are harder to exploit, and the relevance of actual concepts plays a more critical role. For the OOD split, random concepts fail across all concept sizes, highlighting the inability of random concepts to generalize beyond the training distribution. These findings indicate that although random concepts can achieve reasonable performance with a sufficiently large concept set under in-distribution data, they fail when the concept set is limited or when distribution shifts occur. This leaves little opportunity for information leakage, suggesting that the model's performance reliably reflects both the relevance of concepts and the faithfulness of explanations.

## 4.5 CASE STUDY

We investigate how **GCB** explains model predictions through a case study. For each dataset, we sample test instances that are predicted to belong to each class and examine the corresponding concept activation vectors. This allows us to analyze which concepts are (in)active in relation to the

Class 2: Operating systems  Class 3: Computer architecture  Class 4: Computer security

Figure 4: The average concept activations of 10 sampled instances per class across all selected concepts ($K = 30$) as word clouds on `WikiCS`.

predicted labels. Specifically, we visualize the average concept activations of 10 sampled instances per class across all selected concepts ($K = 30$) as word clouds. Figure 4 presents the word clouds for three classes from `WikiCS`, where concepts like "Live USB," "Baikal CPU," and "Encryption" are prominently activated for three different predicted classes. We also use Sankey diagrams to visualize the concept activations for three classes in Figure 15, showing how the model distinguishes between different classes. The complete set of word clouds and Sankey diagrams for all datasets is provided in E.2. They illustrate that the concept activations provide an intuitive and class-discriminative explanation of the model's decision-making process.

## 5 FURTHER DISCUSSION

***GCB vs. LLM-as-predictor methods.*** While some LLM-as-predictor approaches Wang et al. (2024); Chen et al. (2024); Tang et al. (2024) can produce predictions accompanied by natural language explanations that may appear more informative than those from concept bottleneck models, they are fundamentally different. (1) Their explanations are inherently post-hoc: the generated text is not guaranteed to faithfully reflect the actual reasoning process, and how these explanations are produced remains another black box. In contrast, **GCB** makes predictions directly based on the semantics of human-interpretable concepts, ensuring that explanations are faithful by construction and intrinsically aligned with the model's decision process. (2) **GCB** requires access to LLMs only during training. At inference time, no LLM queries are needed. In comparison, LLM-as-predictor methods rely on querying the LLM for each prediction, which incurs substantial computational and monetary costs.

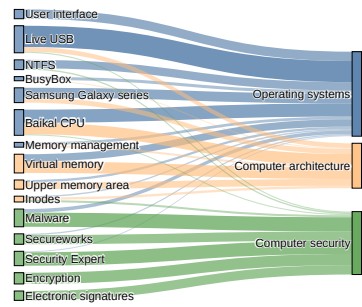

Figure 5: Sankey diagram for `WikiCS` (partial classes).

***GCB's applicability on different graph types.*** The performance of **GCB** largely depends on the quality of the proposed concept space and the effectiveness of the graph-concept alignment model—both of which rely on LLMs for semantic understanding and reasoning over graph instances. To date, LLM-based graph reasoning has primarily focused on text-attributed graphs, which motivates our choice of such graphs as the starting point for exploring **GCB**. Nevertheless, we argue that **GCB** holds strong potential for broader applicability to diverse graph types, such as molecular and biomedical graphs, provided that suitable LLM-driven interfaces Wang et al. (2025); Lee et al. (2025a); Bran et al. (2023) are available to bridge domain-specific graph structures with high-level concepts. We plan to explore this direction as part of our future work.

## 6 CONCLUSION

We present **GCB** as a novel solution for interpretable and robust graph learning. **GCB** maps graph inputs into a human-interpretable concept space, where each concept is expressed in natural language and carries clear semantics. Predictions are then made directly based on these concepts. We conduct extensive experiments and case studies on five real-world datasets from diverse domains, each with distinct challenges, to demonstrate the effectiveness of **GCB**.

## ETHICS STATEMENT

We acknowledge that we have read and adhered to the ICLR Code of Ethics in the preparation and presentation of this work. In line with the principles of responsible stewardship, we are committed to upholding high standards of scientific excellence, honesty, and transparency in our research. We have conducted and presented this work with integrity, giving proper acknowledgment to the contributions of others and ensuring that our findings are reported accurately and reproducibly. We recognize the importance of minimizing potential harms and have reflected on the broader societal impacts of our research, including implications for human well-being and the natural environment. Consistent with the values of fairness and inclusivity, we support the equitable participation of all individuals in research and seek to promote accessibility and inclusiveness in both our methods and outcomes. We further respect the privacy and confidentiality of data that inform scientific discovery, and we endeavor to ensure that our work contributes positively to society, advances knowledge responsibly, and aligns with the long-term public good. At present, we do not identify any specific ethical concerns associated with this research.

## REPRODUCIBILITY STATEMENT

We have taken several steps to ensure the reproducibility of our work. Details of the proposed method (Section 3 and Section B) and training procedures (Section 3 and Section 4.1) are provided, with additional implementation details (Section C) and complete results (Section 4 and Section E) included. A complete description of the data processing steps is also provided in the supplementary materials. Furthermore, we supply anonymized source code in the supplementary materials to facilitate replication of our experiments.

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

APPENDIX

# A  ADDITIONAL RELATED WORK

## A.1  CONCEPT BOTTLENECK MODEL

Concept Bottleneck Models (CBMs) aim to improve model transparency by first mapping inputs into an interpretable set of human-defined concepts (the concept bottleneck), and then making predictions based on those concepts. The original CBM framework Koh et al. (2020) is trained on datasets where each input is annotated with both class labels and corresponding concept labels. At test time, the model predicts concepts from the input and uses them as intermediate representations to produce the final output via a classifier or regressor. This process enhances interpretability and enables human intervention by allowing concept-level edits. However, original CBM Koh et al. (2020) requires substantial human effort to define the concept space and annotate each training sample with concept labels, which can be both time-consuming and labor-intensive. Moreover, they often suffer from suboptimal predictive performance. To address these limitations, Yuksekgonul et al. Yuksekgonul et al. (2023) propose a post-hoc CBM that converts any pretrained model into a concept bottleneck model. Their approach leverages multimodal approaches such as CLIP Radford et al. (2021) to align the input space (e.g., images) with a concept space (e.g., text), thereby reducing the need for explicitly labeled concept data. Nevertheless, this method still requires human expertise or additional learning steps to define the concept subspace. In a concurrent work, Oikarinen et al. Oikarinen et al. (2023) build upon similar ideas but go further by proposing a label-free CBM. They also utilize CLIP's image and text encoders to map inputs to concepts, while fully automating the construction of the concept space using large language models (LLMs). Both approaches Oikarinen et al. (2023); Yuksekgonul et al. (2023) report maintaining competitive predictive performance while improving interpretability.

In addition to these, several works explore specific challenges and extended settings of CBMs. Shang et al. Shang et al. (2024) address the concept completeness problem by proposing to recover missing concepts through transforming complemented vectors with unclear semantics into potential concepts. Shin et al. Shin et al. (2023) conduct in-depth analyses of intervention strategies in CBMs; for instance, they investigate which concept selection criteria are most cost-efficient yet effective in improving task performance. Kim et al. Kim et al. (2023) propose a probabilistic Concept Bottleneck Model to tackle ambiguity in concept prediction, which can undermine model reliability. Their approach explicitly models uncertainty in the concept space and provides explanations incorporating both the predicted concepts and their associated uncertainties. It is also worth mentioning that Xu et al. Xu et al. (2025) introduce a Graph Concept Bottleneck Model that facilitates the modeling of concept relationships by constructing a graph of latent concepts. Although it shares a similar name with our model, it tackles fundamentally different challenges. All of the aforementioned works focus on Euclidean input spaces such as images, and how to adapt Concept Bottleneck Models to graph data remains largely unexplored.

# B  ADDITIONAL DETAILS ON METHODOLOGY

## B.1  PROMPT DETAILS

---

**Prompt for self-supervised concept annotations**

```
Given {graphML} and {dataset-details}.
   1. Provide summary and context analysis on the graph.
   2. Identify a list of key concepts and themes presented in
      the graph.
```

GraphML refers to the graph markup language used for describing the graph (or ego-net if the instance is a node). dataset-details provides a detailed description of the graph dataset, including what each node/edge represents and relevant contextual information.

---

---

**Prompt for Global Concept Proposal**

```
In the domain of {dataset-domain}, list the related
concepts/keywords for classifying the item as {category}.
```

`Dataset-domain` briefly describes the dataset's domain or context, and `category` is the name of a class label from the downstream classification task. We apply this prompt to each class label and aggregate the generated concepts to form the initial concept pool.

---

**Prompt for Instance-Based Concept Extraction**

```
Given a {graphML} and {dataset-details}.
    1. Provide summary and context analysis on the graph.
    2. Identify a list of key concepts presented in the
       graph that are most important for determining its
       classification within the {dataset-domain}, which
       includes the following categories:  {category-list}.
```

`GraphML` refers to the graph markup language used for describing the graph (or ego-net if the instance is a node). `dataset-details` provides a detailed description of the graph dataset, `dataset-domain` briefly describes the dataset's domain or context. `category-list` is the complete list of categories for the classification task to guide the LLM toward generating concepts that are helpful in predicting class labels. Only the outputted concept list from the second step is collected.

---

## B.2 DETAILED PROCEDURES

**Instance-Based Concept Extraction**. We sample $m$ graph instances from each class and apply the prompt to each sampled graph instance, resulting in a large set of candidate concepts. We then identify a subset of concepts that are highly relevant to each class, distinct from those used by other classes, and useful for improving class discrimination. Specifically, for each class $y$, we calculate the class-wise concept activation score as:

$$\bar{C}_y = \frac{1}{|\mathcal{D}_y|} \sum_{x_i \in \mathcal{D}_y} C_i, \tag{6}$$

where $\mathcal{D}_y$ denotes the set of instances belonging to class $y$, and $C_i$ is the concept activation vector for instance $x_i$. Each element $C_i^{(j)}$ represents the activation score (e.g., cosine similarity) between the instance representation $f_\theta^{\text{GNN}}(x_i)$ and the embedding of the $j$-th concept $f^{\text{LM}}(c_j)$.

We then compute the discriminative score of concept $j$ for class $y$ as:

$$\text{score}_j(y) = \bar{C}_y^{(j)} - \frac{1}{|\mathcal{Y}| - 1} \sum_{y' \neq y} \bar{C}_{y'}^{(j)}, \tag{7}$$

where $\mathcal{Y}$ is the set of all class labels, and $\bar{C}_y^{(j)}$ denotes the average activation of concept $j$ for class $y$.

Finally, for each class, we select the top-$k$ concepts with the highest discriminative scores:

$$\mathcal{C}^{\text{inst}} = \text{Top-}k_j(\text{score}_j(y)). \tag{8}$$

**Details of Concept Filtering Process.** Following similar procedures to Oikarinen et al. (2023), we apply a post-processing pipeline to refine the set of candidate concepts. The pipeline consists of the following steps:

*(1) Removing overly long concepts.* Long concepts may reduce both interpretability and generalizability. We therefore tokenize each concept and discard those containing more than 10 tokens.

*(2) Removing concepts overly similar to class labels.* Concepts that are identical or highly similar to class labels undermine the purpose of explanation. To mitigate this issue, we compute the cosine

Table 3: Summary statistics of source and target datasets.

| Dataset | #Nodes | #Edges | Type | Domain | #Class |
|---|---|---|---|---|---|
| Computers | 87,229 | 721,081 | Co-purchase | E-commerce | 10 |
| PubMed | 19,717 | 44,338 | Citation | Biomedicine | 3 |
| Books-History | 41,551 | 358,574 | Co-purchase | E-commerce | 12 |
| Books-Children | 76,875 | 1,554,578 | Co-purchase | E-commerce | 24 |
| Sports-Fitness | 173,055 | 1,773,500 | Co-purchase | E-commerce | 13 |
| Cora | 2,708 | 5,429 | Citation | Computer Science | 7 |
| CiteSeer | 3,186 | 4,277 | Citation | Computer Science | 6 |
| Instagram | 11,339 | 144,010 | User-Post | Social Media | 2 |
| Reddit | 33,434 | 198,448 | Post-Comment | Social Media | 2 |
| WikiCS | 11,701 | 215,863 | Article Link | Wikipedia | 10 |

similarity between the Sentence-BERT embeddings of each concept and each class label, and filter out any concept with a similarity score greater than 0.85.

*(3) Removing redundant concepts.* To reduce redundancy, we compute pairwise cosine similarity among concepts and remove any concept whose similarity with a retained concept exceeds 0.85.

## C  Supplemental experiment setups

### C.1  Details of the datasets

In this section, we summarize the basic statistics of the datasets in our experimental evaluation in Table 3. All datasets used in our study are publicly available and come from diverse domains, including social media networks, citation graphs, and e-commerce graphs. Each node is associated with a class label, and most datasets contain more than two classes. The class distributions are imbalanced in these datasets. Therefore, in our experiments, we report node classification performance using the (Macro-)F1 score and balanced accuracy (BACC).

### C.2  Implementation details

For all GNN-based methods, including those that use GNNs as backbones, we set the hidden dimension to 64 and the number of GNN layers to 2. For GAT and GT models, we use 4 attention heads. For SEGNN Dai & Wang (2021), the original implementation requires access to all training nodes at test time in order to identify the closest neighbors and make predictions based on their labels. However, this approach is incompatible with our inductive setting, where the model is not permitted to access training instances during inference. To address this, we modify the implementation by introducing a small memory buffer that stores 5 representative nodes per class from the training set. During testing, the model is restricted to retrieving neighbors only from this buffer. For all self-explainable graph learning baselines, we follow the default hyperparameter settings provided in their open-source implementations. All experiments are conducted on four NVIDIA L40S GPUs. We access GPT-3.5 via the OpenAI API and set the temperature to 0 during graph summary and concept generation to avoid randomness.

## D  Complexity analysis

The primary overhead of **GCB** lies in the pretraining stage, where a graph encoder is aligned with a semantically meaningful concept space using large language models (LLMs). However, this pretraining is performed once and can be reused across downstream datasets without incurring additional cost. During the main training phase, where we optimize the nformation bottleneck criteria, the dominant cost comes from computing the gate vector $g$ (via a lightweight MLP) and training the classifier $\text{MLP}^{\text{cls}}$ on the masked concept representations. This results in a per-step complexity of $O(BKH)$, where $B$ is the batch size, $K$ is the number of candidate concepts, and $H$ is the hidden dimension of the MLP. The final predictor, after concept selection, operates on a reduced concept set and is simply a small MLP, which is highly efficient in both training and inference. At inference time, **GCB** consists of a frozen graph encoder (e.g., a GNN) followed by a fixed MLP classifier over selected concepts, making its runtime complexity comparable to that of a standard GNN model.

Table 4: Node classification performance in *regular settings*. The best-performing interpretable GNN on each dataset is underlined, and the overall best-performing method is **bolded**.

| Method | Cora | | Citeseer | | Instagram | | Reddit | | WikiCS | |
|---|---|---|---|---|---|---|---|---|---|---|
| | F1 (%) | BACC (%) | F1 (%) | BACC (%) | F1 (%) | BACC (%) | F1 (%) | BACC (%) | F1 (%) | BACC (%) |
| MLP | $68.00_{(0.89)}$ | $67.49_{(0.76)}$ | $63.81_{(0.37)}$ | $64.29_{(0.34)}$ | $53.78_{(0.59)}$ | $53.76_{(0.58)}$ | $53.34_{(0.77)}$ | $53.39_{(0.76)}$ | $69.22_{(0.51)}$ | $69.25_{(0.57)}$ |
| GCN | $72.38_{(0.58)}$ | $71.95_{(0.49)}$ | $62.47_{(0.29)}$ | $63.05_{(0.33)}$ | $53.63_{(0.84)}$ | $53.62_{(0.82)}$ | $54.49_{(1.19)}$ | $54.64_{(1.03)}$ | $67.45_{(1.76)}$ | $68.84_{(1.82)}$ |
| GAT | **$74.58_{(0.95)}$** | **$74.42_{(0.90)}$** | $63.92_{(0.92)}$ | $64.47_{(0.91)}$ | $55.71_{(1.06)}$ | $55.74_{(1.10)}$ | **$56.29_{(0.60)}$** | $56.30_{(0.59)}$ | $67.54_{(1.68)}$ | $68.14_{(1.59)}$ |
| SAGE | $70.59_{(0.68)}$ | $70.66_{(0.80)}$ | **$65.09_{(0.62)}$** | **$65.52_{(0.64)}$** | $54.49_{(0.46)}$ | $54.50_{(0.46)}$ | $55.33_{(0.38)}$ | $55.33_{(0.38)}$ | **$72.96_{(0.30)}$** | **$72.74_{(0.40)}$** |
| GT | $72.36_{(1.96)}$ | $72.18_{(1.56)}$ | $64.40_{(0.76)}$ | $64.93_{(0.68)}$ | $54.79_{(0.40)}$ | $54.77_{(0.39)}$ | $56.15_{(0.39)}$ | $56.15_{(0.39)}$ | $72.27_{(0.52)}$ | $72.46_{(0.56)}$ |
| DIR-GNN | $73.03_{(2.62)}$ | $72.51_{(1.90)}$ | $62.10_{(0.58)}$ | $64.67_{(0.50)}$ | **$56.76_{(1.24)}$** | **$57.37_{(0.95)}$** | $55.34_{(1.81)}$ | **$57.18_{(0.48)}$** | $67.14_{(3.60)}$ | $66.26_{(3.83)}$ |
| GIB | $66.81_{(4.23)}$ | $67.23_{(4.02)}$ | $49.28_{(14.03)}$ | $53.88_{(11.42)}$ | $40.72_{(8.44)}$ | $51.52_{(1.86)}$ | $38.84_{(8.18)}$ | $51.49_{(2.11)}$ | $45.30_{(18.50)}$ | $45.38_{(14.85)}$ |
| VGIB | $63.46_{(28.19)}$ | $64.59_{(25.11)}$ | $53.90_{(19.24)}$ | $56.99_{(16.88)}$ | $39.64_{(1.64)}$ | $50.29_{(0.58)}$ | $33.68_{(1.13)}$ | $50.12_{(0.22)}$ | $61.44_{(25.27)}$ | $62.90_{(22.46)}$ |
| SEGNN | $49.90_{(4.09)}$ | $53.07_{(3.30)}$ | $52.12_{(5.51)}$ | $55.67_{(4.11)}$ | $44.71_{(2.56)}$ | $51.04_{(0.49)}$ | $53.53_{(1.66)}$ | $54.59_{(0.90)}$ | $28.87_{(3.57)}$ | $34.71_{(2.78)}$ |
| **GCB** | $70.54_{(1.33)}$ | $71.41_{(0.88)}$ | $63.22_{(0.50)}$ | $63.54_{(0.49)}$ | **$56.76_{(0.55)}$** | $56.71_{(0.51)}$ | $55.06_{(0.72)}$ | $55.11_{(0.72)}$ | $68.82_{(0.41)}$ | $70.64_{(0.82)}$ |

Table 5: Node classification performance in *OOD settings* with upsampling ratio $\gamma = 2$. The best-performing interpretable GNN on each dataset is underlined, and the overall best-performing method is **bolded**.

| Method | Cora | | Citeseer | | Instagram | | Reddit | | WikiCS | |
|---|---|---|---|---|---|---|---|---|---|---|
| | F1 (%) | BACC (%) | F1 (%) | BACC (%) | F1 (%) | BACC (%) | F1 (%) | BACC (%) | F1 (%) | BACC (%) |
| MLP | $46.52_{(0.59)}$ | $57.50_{(0.69)}$ | $44.89_{(0.70)}$ | $60.49_{(0.77)}$ | $35.55_{(0.59)}$ | $51.54_{(0.20)}$ | $17.03_{(0.51)}$ | $51.28_{(0.56)}$ | $53.82_{(0.42)}$ | $63.23_{(0.40)}$ |
| GCN | **$56.10_{(0.46)}$** | **$64.04_{(0.52)}$** | $41.06_{(0.43)}$ | $56.08_{(0.57)}$ | $39.36_{(3.25)}$ | $52.54_{(0.86)}$ | $16.65_{(0.73)}$ | $50.74_{(0.20)}$ | $53.80_{(0.55)}$ | $60.37_{(1.37)}$ |
| GAT | $52.54_{(1.48)}$ | $63.29_{(1.31)}$ | $44.71_{(0.68)}$ | $60.63_{(0.56)}$ | $33.42_{(0.37)}$ | $51.49_{(0.12)}$ | $13.06_{(0.28)}$ | $49.78_{(0.39)}$ | **$56.41_{(2.24)}$** | $64.32_{(1.67)}$ |
| SAGE | $40.39_{(1.19)}$ | $50.69_{(1.08)}$ | $40.99_{(0.99)}$ | $56.02_{(0.78)}$ | $35.71_{(0.42)}$ | $51.76_{(0.31)}$ | $15.97_{(0.35)}$ | $50.74_{(0.51)}$ | $49.65_{(0.64)}$ | $60.20_{(0.83)}$ |
| GT | $42.83_{(1.35)}$ | $51.62_{(1.59)}$ | $40.57_{(1.22)}$ | $56.93_{(0.90)}$ | $33.83_{(0.60)}$ | $51.47_{(0.28)}$ | $13.22_{(0.33)}$ | $50.14_{(0.34)}$ | $51.10_{(0.69)}$ | $59.51_{(0.99)}$ |
| DIR-GNN | $18.54_{(2.90)}$ | $40.48_{(2.24)}$ | $15.18_{(0.70)}$ | $42.44_{(0.52)}$ | $26.74_{(0.00)}$ | $50.00_{(0.00)}$ | $8.46_{(0.00)}$ | $50.00_{(0.00)}$ | $23.41_{(1.92)}$ | $42.89_{(0.55)}$ |
| GIB | $21.45_{(3.55)}$ | $40.93_{(1.77)}$ | $16.98_{(4.91)}$ | $43.81_{(3.24)}$ | $26.76_{(0.02)}$ | $50.01_{(0.01)}$ | $8.53_{(0.06)}$ | $49.99_{(0.06)}$ | $23.93_{(1.12)}$ | $41.10_{(1.63)}$ |
| VGIB | $45.60_{(4.00)}$ | $58.19_{(2.70)}$ | $15.61_{(1.75)}$ | $44.07_{(1.00)}$ | $26.74_{(0.00)}$ | $50.00_{(0.00)}$ | $8.46_{(0.00)}$ | $50.00_{(0.00)}$ | $54.31_{(0.90)}$ | $63.54_{(0.66)}$ |
| SEGNN | $40.04_{(2.47)}$ | $51.44_{(2.36)}$ | $25.59_{(2.76)}$ | $45.69_{(1.58)}$ | $26.74_{(0.00)}$ | $50.00_{(0.00)}$ | $8.46_{(0.00)}$ | $50.00_{(0.00)}$ | $37.26_{(1.18)}$ | $49.80_{(0.91)}$ |
| **GCB** | $54.23_{(0.00)}$ | $62.97_{(1.15)}$ | **$57.46_{(0.85)}$** | **$65.48_{(0.65)}$** | **$53.20_{(0.81)}$** | **$55.89_{(0.82)}$** | **$43.74_{(0.77)}$** | **$57.99_{(0.71)}$** | $55.19_{(0.72)}$ | **$66.36_{(0.62)}$** |

# E ADDITIONAL RESULTS

## E.1 ROBUSTNESS EVALUATION

We report the results for all additional upsampling ratios $\gamma \in \{2, 3, 10\}$ in Table 5, Table 6, and Table 7, respectively. Results for different perturbation ratios $\rho \in \{0.05, 0.1, 0.2, 0.5\}$ are shown in Table 8, Table 9, Table 10, and Table 11.

We emphasize that the test splits used for different upsampling ratios are not aligned, making direct comparison across these settings inappropriate. While a larger upsampling ratio increases the distribution shift between the training and test sets, it may also lead to a more balanced class distribution in the training or test data, which can sometimes improve test performance. Regarding the perturbation setting, we observe that **GCB** is the least affected by structural perturbation. We attribute this to the use of a fixed pretrained encoder, which is not updated during task-specific training. As a result, perturbing the training graph does not alter the graph embedding function. Moreover, the data augmentation used during pretraining also contributes to **GCB**'s robustness under structural noise. Interestingly, across all baseline methods, we do not observe a consistent trend correlating performance with increasing perturbation ratio. One possible explanation is that, for perturbation-sensitive models, even a small perturbation (e.g., $\rho = 0.05$) significantly degrades performance, and the marginal impact of further perturbation is limited. Furthermore, recent studies such as Han et al. (2023) have shown that some GNNs can perform well even when trained without graph structure—effectively functioning like MLPs—and still generalize well when tested with full graph connectivity. When the perturbation ratio is large, models may similarly learn to disregard noisy structure, exhibiting behavior consistent with such MLP-based approaches and mitigating the negative effects of edge pertubation.

Table 6: Node classification performance in *OOD settings* with upsampling ratio $\gamma = 3$. The best-performing interpretable GNN on each dataset is underlined, and the overall best-performing method is **bolded**.

| Method | Cora F1 (%) | Cora BACC (%) | Citeseer F1 (%) | Citeseer BACC (%) | Instagram F1 (%) | Instagram BACC (%) | Reddit F1 (%) | Reddit BACC (%) | WikiCS F1 (%) | WikiCS BACC (%) |
|---|---|---|---|---|---|---|---|---|---|---|
| MLP | $43.78_{(0.52)}$ | $54.85_{(0.57)}$ | $45.73_{(0.59)}$ | $58.72_{(0.71)}$ | $37.04_{(0.67)}$ | $52.33_{(0.25)}$ | $16.37_{(0.61)}$ | $51.37_{(0.32)}$ | $55.01_{(0.33)}$ | $65.33_{(1.15)}$ |
| GCN | $\mathbf{57.28}_{(1.35)}$ | $\mathbf{67.86}_{(0.98)}$ | $41.59_{(0.68)}$ | $56.90_{(0.80)}$ | $37.73_{(3.06)}$ | $51.58_{(0.59)}$ | $16.00_{(0.35)}$ | $49.23_{(0.48)}$ | $54.99_{(1.28)}$ | $61.65_{(2.33)}$ |
| GAT | $52.81_{(1.43)}$ | $60.66_{(1.28)}$ | $43.09_{(1.45)}$ | $58.16_{(1.39)}$ | $34.77_{(1.08)}$ | $51.50_{(0.38)}$ | $13.85_{(0.44)}$ | $49.44_{(0.28)}$ | $\mathbf{56.99}_{(0.10)}$ | $65.85_{(0.84)}$ |
| SAGE | $51.40_{(1.77)}$ | $62.46_{(1.54)}$ | $36.25_{(1.39)}$ | $52.63_{(1.14)}$ | $35.04_{(0.55)}$ | $51.47_{(0.27)}$ | $14.36_{(0.17)}$ | $48.96_{(0.40)}$ | $52.43_{(0.78)}$ | $60.92_{(0.97)}$ |
| GT | $47.70_{(1.31)}$ | $58.26_{(1.19)}$ | $36.12_{(1.62)}$ | $53.10_{(1.23)}$ | $33.16_{(0.47)}$ | $50.15_{(0.37)}$ | $14.18_{(0.15)}$ | $49.59_{(0.21)}$ | $54.83_{(0.89)}$ | $62.58_{(1.07)}$ |
| DIR-GNN | $20.13_{(2.75)}$ | $41.89_{(1.50)}$ | $14.70_{(0.34)}$ | $43.15_{(0.38)}$ | $26.74_{(0.00)}$ | $50.00_{(0.00)}$ | $8.46_{(0.00)}$ | $50.00_{(0.00)}$ | $23.60_{(1.40)}$ | $42.84_{(0.57)}$ |
| GIB | $22.43_{(5.91)}$ | $41.53_{(4.24)}$ | $17.72_{(5.88)}$ | $44.14_{(4.31)}$ | $26.74_{(0.00)}$ | $50.00_{(0.01)}$ | $8.48_{(0.04)}$ | $50.01_{(0.02)}$ | $20.30_{(7.70)}$ | $35.16_{(9.59)}$ |
| VGIB | $44.05_{(2.53)}$ | $57.14_{(1.92)}$ | $17.14_{(4.35)}$ | $44.50_{(2.44)}$ | $26.74_{(0.00)}$ | $50.00_{(0.00)}$ | $8.46_{(0.00)}$ | $50.00_{(0.00)}$ | $54.74_{(1.32)}$ | $63.15_{(1.20)}$ |
| SEGNN | $29.92_{(1.05)}$ | $46.62_{(0.81)}$ | $25.15_{(9.17)}$ | $43.70_{(6.98)}$ | $26.74_{(0.00)}$ | $50.00_{(0.00)}$ | $8.46_{(0.00)}$ | $50.00_{(0.00)}$ | $27.85_{(1.02)}$ | $43.33_{(1.50)}$ |
| **GCB** | $\underline{54.99}_{(0.00)}$ | $\underline{65.00}_{(0.00)}$ | $\underline{\mathbf{57.85}}_{(0.27)}$ | $\underline{65.62}_{(0.79)}$ | $\underline{\mathbf{54.54}}_{(0.17)}$ | $\underline{\mathbf{56.39}}_{(0.36)}$ | $\underline{\mathbf{45.82}}_{(0.31)}$ | $\underline{\mathbf{60.65}}_{(0.83)}$ | $\underline{54.97}_{(0.37)}$ | $\underline{\mathbf{66.70}}_{(0.40)}$ |

Table 7: Node classification performance in *OOD settings* with upsampling ratio $\gamma = 10$. The best-performing interpretable GNN on each dataset is underlined, and the overall best-performing method is **bolded**.

| Method | Cora F1 (%) | Cora BACC (%) | Citeseer F1 (%) | Citeseer BACC (%) | Instagram F1 (%) | Instagram BACC (%) | Reddit F1 (%) | Reddit BACC (%) | WikiCS F1 (%) | WikiCS BACC (%) |
|---|---|---|---|---|---|---|---|---|---|---|
| MLP | $47.58_{(0.44)}$ | $59.14_{(0.61)}$ | $41.44_{(0.42)}$ | $56.87_{(0.70)}$ | $35.38_{(0.66)}$ | $51.29_{(0.43)}$ | $16.08_{(0.42)}$ | $50.69_{(0.34)}$ | $52.72_{(0.50)}$ | $62.79_{(0.50)}$ |
| GCN | $\mathbf{62.08}_{(1.59)}$ | $69.26_{(1.83)}$ | $43.58_{(0.45)}$ | $54.79_{(0.34)}$ | $47.15_{(4.10)}$ | $55.23_{(1.48)}$ | $17.35_{(0.97)}$ | $49.91_{(0.17)}$ | $62.47_{(1.15)}$ | $64.89_{(1.38)}$ |
| GAT | $60.32_{(1.56)}$ | $68.94_{(1.15)}$ | $48.46_{(0.66)}$ | $61.74_{(0.80)}$ | $35.80_{(0.67)}$ | $51.70_{(0.30)}$ | $15.35_{(1.35)}$ | $51.40_{(0.38)}$ | $57.17_{(1.59)}$ | $60.82_{(1.69)}$ |
| SAGE | $50.49_{(1.06)}$ | $57.96_{(1.12)}$ | $35.75_{(1.49)}$ | $53.22_{(0.90)}$ | $37.94_{(0.94)}$ | $52.14_{(0.40)}$ | $16.70_{(0.73)}$ | $52.08_{(0.22)}$ | $\mathbf{62.88}_{(0.18)}$ | $\mathbf{72.40}_{(1.02)}$ |
| GT | $47.31_{(2.61)}$ | $56.27_{(1.92)}$ | $30.80_{(1.22)}$ | $50.68_{(0.82)}$ | $37.51_{(0.46)}$ | $52.90_{(0.16)}$ | $17.33_{(0.79)}$ | $51.58_{(0.24)}$ | $62.18_{(0.80)}$ | $68.79_{(1.87)}$ |
| DIR-GNN | $22.04_{(3.39)}$ | $42.60_{(2.67)}$ | $15.56_{(1.05)}$ | $42.93_{(0.37)}$ | $26.74_{(0.00)}$ | $50.00_{(0.00)}$ | $8.46_{(0.00)}$ | $50.00_{(0.00)}$ | $23.89_{(0.58)}$ | $42.04_{(0.65)}$ |
| GIB | $26.30_{(8.89)}$ | $44.06_{(5.96)}$ | $14.94_{(0.54)}$ | $42.42_{(0.68)}$ | $26.92_{(0.31)}$ | $50.06_{(0.13)}$ | $8.46_{(0.02)}$ | $49.98_{(0.05)}$ | $25.39_{(2.25)}$ | $38.19_{(1.06)}$ |
| VGIB | $60.87_{(3.20)}$ | $69.42_{(3.04)}$ | $24.29_{(6.80)}$ | $48.20_{(3.96)}$ | $26.74_{(0.00)}$ | $50.00_{(0.00)}$ | $8.46_{(0.00)}$ | $50.00_{(0.00)}$ | $61.85_{(1.81)}$ | $69.02_{(1.35)}$ |
| **GCB** | $\underline{61.68}_{(1.63)}$ | $\underline{\mathbf{69.55}}_{(1.11)}$ | $\underline{\mathbf{58.08}}_{(0.34)}$ | $\underline{\mathbf{65.52}}_{(0.25)}$ | $\underline{\mathbf{52.17}}_{(2.34)}$ | $\underline{\mathbf{55.57}}_{(1.15)}$ | $\underline{\mathbf{44.77}}_{(1.34)}$ | $\underline{\mathbf{55.75}}_{(0.43)}$ | $\underline{60.66}_{(0.97)}$ | $\underline{71.22}_{(1.43)}$ |

Table 8: Node classification performance in *adversarial settings* with perturbation ratio $\rho = 0.05$. The best-performing interpretable GNN on each dataset is underlined, and the overall best-performing method is **bolded**.

| Method | Cora F1 (%) | Cora BACC (%) | Citeseer F1 (%) | Citeseer BACC (%) | Instagram F1 (%) | Instagram BACC (%) | Reddit F1 (%) | Reddit BACC (%) | WikiCS F1 (%) | WikiCS BACC (%) |
|---|---|---|---|---|---|---|---|---|---|---|
| MLP | $46.79_{(0.88)}$ | $58.83_{(0.84)}$ | $38.16_{(0.41)}$ | $53.71_{(0.37)}$ | $37.69_{(0.44)}$ | $52.53_{(0.15)}$ | $16.19_{(0.46)}$ | $51.58_{(0.44)}$ | $53.96_{(0.34)}$ | $64.83_{(0.25)}$ |
| GCN | $58.93_{(1.32)}$ | $67.16_{(1.38)}$ | $46.96_{(0.95)}$ | $58.48_{(1.13)}$ | $42.47_{(4.66)}$ | $52.11_{(1.09)}$ | $15.95_{(0.86)}$ | $50.46_{(0.68)}$ | $62.44_{(0.37)}$ | $68.39_{(0.49)}$ |
| GAT | $55.01_{(1.93)}$ | $61.57_{(1.94)}$ | $43.59_{(1.57)}$ | $57.78_{(1.21)}$ | $34.32_{(1.37)}$ | $51.16_{(0.53)}$ | $17.03_{(0.95)}$ | $51.28_{(0.42)}$ | $58.93_{(2.81)}$ | $66.43_{(2.35)}$ |
| SAGE | $53.45_{(2.23)}$ | $57.21_{(2.34)}$ | $42.45_{(1.44)}$ | $57.74_{(0.95)}$ | $40.93_{(6.08)}$ | $52.32_{(0.67)}$ | $16.00_{(0.61)}$ | $51.42_{(0.61)}$ | $61.58_{(0.22)}$ | $68.96_{(1.30)}$ |
| GT | $38.25_{(2.04)}$ | $45.19_{(1.18)}$ | $39.80_{(3.32)}$ | $55.30_{(2.16)}$ | $34.43_{(1.15)}$ | $51.44_{(0.30)}$ | $14.35_{(0.74)}$ | $51.02_{(0.27)}$ | $56.30_{(1.16)}$ | $64.35_{(1.49)}$ |
| DIR-GNN | $\underline{\mathbf{73.48}}_{(1.08)}$ | $\underline{\mathbf{72.72}}_{(1.37)}$ | $62.03_{(0.64)}$ | $\underline{\mathbf{64.60}}_{(0.54)}$ | $55.78_{(2.54)}$ | $56.70_{(1.42)}$ | $54.64_{(2.70)}$ | $57.12_{(1.00)}$ | $65.05_{(1.45)}$ | $63.77_{(1.50)}$ |
| GIB | $58.60_{(15.18)}$ | $59.17_{(14.38)}$ | $45.60_{(17.26)}$ | $50.91_{(13.49)}$ | $40.96_{(8.68)}$ | $51.59_{(1.93)}$ | $38.57_{(7.79)}$ | $51.70_{(2.56)}$ | $40.07_{(16.67)}$ | $40.14_{(12.96)}$ |
| VGIB | $21.17_{(26.63)}$ | $26.65_{(23.68)}$ | $53.90_{(19.09)}$ | $57.17_{(16.80)}$ | $38.99_{(0.34)}$ | $50.07_{(0.14)}$ | $34.58_{(2.56)}$ | $50.29_{(0.47)}$ | $\underline{\mathbf{72.78}}_{(1.07)}$ | $\underline{\mathbf{72.45}}_{(1.37)}$ |
| SEGNN | $55.79_{(1.48)}$ | $59.39_{(1.03)}$ | $60.06_{(0.69)}$ | $62.95_{(0.74)}$ | $54.75_{(1.01)}$ | $55.22_{(0.91)}$ | $55.58_{(0.36)}$ | $\underline{\mathbf{55.99}}_{(0.30)}$ | $37.35_{(0.71)}$ | $41.85_{(1.12)}$ |
| **GCB** | $70.75_{(0.85)}$ | $71.34_{(1.05)}$ | $\underline{\mathbf{63.20}}_{(0.76)}$ | $63.52_{(0.78)}$ | $\underline{\mathbf{56.79}}_{(0.60)}$ | $\underline{\mathbf{56.72}}_{(0.59)}$ | $\underline{54.93}_{(0.78)}$ | $54.98_{(0.79)}$ | $68.70_{(0.42)}$ | $70.60_{(0.46)}$ |

Table 9: Node classification performance in *adversarial settings* with perturbation ratio $\rho = 0.1$. The best-performing interpretable GNN on each dataset is underlined, and the overall best-performing method is **bolded**.

| Method | Cora | | Citeseer | | Instagram | | Reddit | | WikiCS | |
|---|---|---|---|---|---|---|---|---|---|---|
| | F1 (%) | BACC (%) | F1 (%) | BACC (%) | F1 (%) | BACC (%) | F1 (%) | BACC (%) | F1 (%) | BACC (%) |
| MLP | $45.04_{(1.20)}$ | $57.94_{(0.91)}$ | $41.94_{(0.33)}$ | $57.38_{(0.21)}$ | $34.65_{(0.60)}$ | $51.57_{(0.30)}$ | $18.09_{(0.43)}$ | $51.09_{(0.58)}$ | $54.13_{(0.30)}$ | $64.72_{(0.64)}$ |
| GCN | $65.19_{(1.66)}$ | $67.62_{(1.61)}$ | $46.43_{(1.13)}$ | $58.40_{(1.17)}$ | $38.56_{(1.38)}$ | $51.96_{(0.60)}$ | $18.17_{(1.20)}$ | $50.51_{(0.67)}$ | $63.15_{(2.44)}$ | $68.98_{(2.14)}$ |
| GAT | $63.56_{(1.59)}$ | $68.88_{(1.40)}$ | $43.79_{(1.41)}$ | $58.11_{(0.90)}$ | $35.94_{(2.11)}$ | $51.83_{(0.55)}$ | $17.30_{(1.53)}$ | $51.83_{(0.72)}$ | $58.79_{(1.66)}$ | $67.83_{(1.26)}$ |
| SAGE | $47.78_{(0.92)}$ | $55.17_{(0.52)}$ | $40.60_{(0.80)}$ | $56.65_{(0.63)}$ | $38.86_{(1.28)}$ | $52.61_{(0.62)}$ | $16.93_{(0.48)}$ | $51.67_{(0.34)}$ | $59.90_{(0.67)}$ | $66.69_{(0.61)}$ |
| GT | $40.32_{(1.93)}$ | $46.77_{(1.27)}$ | $27.14_{(1.44)}$ | $46.19_{(1.04)}$ | $35.26_{(0.73)}$ | $51.95_{(0.35)}$ | $16.80_{(1.02)}$ | $51.26_{(0.49)}$ | $60.67_{(0.99)}$ | $67.86_{(1.05)}$ |
| DIR-GNN | $71.70_{(2.79)}$ | $71.04_{(2.08)}$ | $61.84_{(1.36)}$ | **$64.42_{(1.24)}$** | $55.55_{(2.17)}$ | $56.66_{(1.28)}$ | **$55.41_{(1.29)}$** | **$57.48_{(0.53)}$** | $64.30_{(4.20)}$ | $63.15_{(4.09)}$ |
| GIB | $55.55_{(16.26)}$ | $58.38_{(11.63)}$ | $58.99_{(4.11)}$ | $61.91_{(3.36)}$ | $41.53_{(9.29)}$ | $51.81_{(2.20)}$ | $40.23_{(8.43)}$ | $52.11_{(2.82)}$ | $30.36_{(14.24)}$ | $32.54_{(12.21)}$ |
| VGIB | $22.42_{(26.34)}$ | $28.23_{(23.26)}$ | $52.91_{(22.73)}$ | $55.81_{(19.19)}$ | $38.86_{(0.07)}$ | $50.02_{(0.03)}$ | $33.92_{(1.58)}$ | $50.16_{(0.30)}$ | $59.83_{(24.76)}$ | $60.32_{(22.57)}$ |
| SEGNN | $56.89_{(0.75)}$ | $60.23_{(0.64)}$ | $59.55_{(0.62)}$ | $62.66_{(0.62)}$ | $54.67_{(0.70)}$ | $55.42_{(0.77)}$ | $55.91_{(1.85)}$ | $56.70_{(1.09)}$ | $36.78_{(1.67)}$ | $41.06_{(1.82)}$ |
| **GCB** | $70.54_{(1.54)}$ | **$71.31_{(2.31)}$** | **$63.02_{(0.40)}$** | $63.38_{(0.44)}$ | **$56.75_{(0.36)}$** | **$56.70_{(0.38)}$** | $54.91_{(0.40)}$ | $54.95_{(0.38)}$ | **$68.80_{(0.30)}$** | **$70.45_{(0.43)}$** |

Table 10: Node classification performance in *adversarial settings* with perturbation ratio $\rho = 0.2$. The best-performing interpretable GNN on each dataset is underlined, and the overall best-performing method is **bolded**.

| Method | Cora | | Citeseer | | Instagram | | Reddit | | WikiCS | |
|---|---|---|---|---|---|---|---|---|---|---|
| | F1 (%) | BACC (%) | F1 (%) | BACC (%) | F1 (%) | BACC (%) | F1 (%) | BACC (%) | F1 (%) | BACC (%) |
| MLP | $49.30_{(0.81)}$ | $59.94_{(0.97)}$ | $42.01_{(0.43)}$ | $57.97_{(0.22)}$ | $37.57_{(3.23)}$ | $51.27_{(0.41)}$ | $15.37_{(0.09)}$ | $51.51_{(0.29)}$ | $52.92_{(0.41)}$ | $61.84_{(0.70)}$ |
| GCN | $60.24_{(0.83)}$ | $68.81_{(1.01)}$ | $47.16_{(1.29)}$ | $58.67_{(1.02)}$ | $37.70_{(2.16)}$ | $51.52_{(0.53)}$ | $17.24_{(1.82)}$ | $50.60_{(0.84)}$ | $59.73_{(0.71)}$ | $62.88_{(0.35)}$ |
| GAT | $57.88_{(2.24)}$ | $64.39_{(1.85)}$ | $44.69_{(1.35)}$ | $58.83_{(1.13)}$ | $37.82_{(1.08)}$ | $52.08_{(0.46)}$ | $14.93_{(1.04)}$ | $50.50_{(0.56)}$ | $58.14_{(2.12)}$ | $62.40_{(2.18)}$ |
| SAGE | $50.26_{(1.95)}$ | $57.82_{(1.43)}$ | $29.81_{(2.27)}$ | $49.99_{(1.55)}$ | $36.98_{(0.47)}$ | $52.00_{(0.08)}$ | $17.10_{(0.40)}$ | $50.99_{(0.43)}$ | $62.87_{(0.63)}$ | $70.36_{(0.31)}$ |
| GT | $51.12_{(2.34)}$ | $56.14_{(2.39)}$ | $32.63_{(1.41)}$ | $51.08_{(0.71)}$ | $35.34_{(0.86)}$ | $51.61_{(0.50)}$ | $16.19_{(0.68)}$ | $50.84_{(0.24)}$ | $60.47_{(0.47)}$ | $66.23_{(0.77)}$ |
| DIR-GNN | $71.30_{(2.36)}$ | **$71.11_{(1.94)}$** | $62.54_{(0.34)}$ | **$65.12_{(0.38)}$** | $55.77_{(2.23)}$ | **$56.87_{(1.32)}$** | $54.68_{(2.36)}$ | **$56.99_{(0.90)}$** | $61.70_{(3.35)}$ | $60.60_{(3.45)}$ |
| GIB | $37.52_{(19.90)}$ | $42.46_{(16.75)}$ | $52.91_{(12.84)}$ | $57.50_{(9.10)}$ | $40.83_{(8.60)}$ | $51.53_{(1.89)}$ | $41.45_{(9.60)}$ | $51.65_{(2.13)}$ | $24.40_{(11.79)}$ | $27.94_{(10.41)}$ |
| VGIB | $34.11_{(32.96)}$ | $38.47_{(29.40)}$ | $52.05_{(22.62)}$ | $55.80_{(19.34)}$ | $40.36_{(3.07)}$ | $50.41_{(0.81)}$ | $33.09_{(0.08)}$ | $50.00_{(0.02)}$ | $59.54_{(24.67)}$ | $61.20_{(21.58)}$ |
| SEGNN | $55.76_{(1.87)}$ | $59.44_{(1.21)}$ | $59.94_{(0.64)}$ | $62.98_{(0.54)}$ | $55.07_{(1.63)}$ | $55.27_{(1.65)}$ | $54.56_{(0.07)}$ | $55.35_{(0.37)}$ | $35.77_{(0.70)}$ | $40.40_{(0.89)}$ |
| **GCB** | $70.40_{(1.32)}$ | $71.03_{(0.60)}$ | **$63.14_{(0.74)}$** | $63.47_{(0.70)}$ | **$56.81_{(0.28)}$** | $56.76_{(0.30)}$ | **$55.05_{(0.44)}$** | $55.10_{(0.45)}$ | **$68.71_{(0.55)}$** | **$70.64_{(0.59)}$** |

Table 11: Node classification performance in *adversarial settings* with perturbation ratio $\rho = 0.5$. The best-performing interpretable GNN on each dataset is underlined, and the overall best-performing method is **bolded**.

| Method | Cora | | Citeseer | | Instagram | | Reddit | | WikiCS | |
|---|---|---|---|---|---|---|---|---|---|---|
| | F1 (%) | BACC (%) | F1 (%) | BACC (%) | F1 (%) | BACC (%) | F1 (%) | BACC (%) | F1 (%) | BACC (%) |
| MLP | $47.76_{(0.43)}$ | $58.52_{(0.43)}$ | $44.65_{(0.39)}$ | $58.45_{(0.41)}$ | $35.26_{(0.57)}$ | $51.72_{(0.39)}$ | $16.78_{(0.31)}$ | $50.46_{(0.59)}$ | $55.24_{(0.16)}$ | $65.36_{(1.10)}$ |
| GCN | $52.19_{(1.00)}$ | $61.01_{(0.95)}$ | $46.07_{(1.23)}$ | $56.61_{(1.12)}$ | $42.37_{(1.34)}$ | $53.07_{(0.85)}$ | $16.20_{(1.08)}$ | $50.73_{(0.48)}$ | $65.21_{(0.63)}$ | $68.76_{(1.02)}$ |
| GAT | $54.25_{(2.86)}$ | $63.55_{(1.63)}$ | $46.58_{(0.47)}$ | $60.72_{(0.81)}$ | $37.55_{(1.31)}$ | $52.63_{(0.43)}$ | $18.61_{(0.96)}$ | $51.83_{(0.37)}$ | $59.33_{(2.20)}$ | $65.94_{(1.80)}$ |
| SAGE | $44.04_{(1.28)}$ | $50.19_{(0.89)}$ | $32.53_{(2.83)}$ | $50.50_{(1.70)}$ | $36.20_{(0.84)}$ | $52.24_{(0.19)}$ | $16.29_{(1.02)}$ | $51.15_{(0.41)}$ | $62.65_{(0.42)}$ | **$70.73_{(0.50)}$** |
| GT | $42.25_{(1.81)}$ | $52.13_{(2.09)}$ | $31.04_{(2.55)}$ | $49.44_{(2.11)}$ | $35.67_{(0.87)}$ | $51.02_{(0.17)}$ | $13.85_{(0.83)}$ | $50.82_{(0.54)}$ | $62.95_{(1.19)}$ | $70.35_{(1.05)}$ |
| DIR-GNN | $71.44_{(0.96)}$ | $69.95_{(1.33)}$ | $62.32_{(0.72)}$ | **$64.97_{(0.63)}$** | $52.83_{(7.05)}$ | $55.64_{(3.03)}$ | $54.60_{(2.34)}$ | $56.17_{(1.02)}$ | $53.83_{(4.77)}$ | $54.12_{(3.56)}$ |
| GIB | $25.59_{(18.05)}$ | $31.80_{(16.51)}$ | $40.56_{(16.48)}$ | $46.87_{(13.35)}$ | $38.11_{(5.95)}$ | $50.56_{(0.69)}$ | $38.93_{(8.60)}$ | $51.73_{(2.61)}$ | $17.64_{(8.56)}$ | $23.25_{(8.04)}$ |
| VGIB | $20.64_{(27.56)}$ | $26.54_{(24.51)}$ | $54.25_{(20.38)}$ | $56.74_{(17.96)}$ | $39.00_{(0.29)}$ | $50.06_{(0.13)}$ | $36.58_{(7.07)}$ | $50.60_{(1.19)}$ | $45.45_{(31.44)}$ | $47.58_{(27.67)}$ |
| SEGNN | $56.47_{(0.72)}$ | $59.51_{(0.94)}$ | $60.23_{(0.68)}$ | $63.10_{(0.72)}$ | $54.32_{(0.52)}$ | $54.61_{(0.75)}$ | $55.81_{(1.45)}$ | **$56.36_{(1.57)}$** | $35.41_{(0.52)}$ | $39.84_{(0.52)}$ |
| **GCB** | $70.48_{(2.31)}$ | **$70.80_{(1.28)}$** | **$63.39_{(0.37)}$** | $63.76_{(0.38)}$ | **$56.95_{(0.18)}$** | **$56.91_{(0.19)}$** | **$55.02_{(0.67)}$** | $55.12_{(0.68)}$ | **$69.17_{(0.45)}$** | $70.45_{(0.51)}$ |

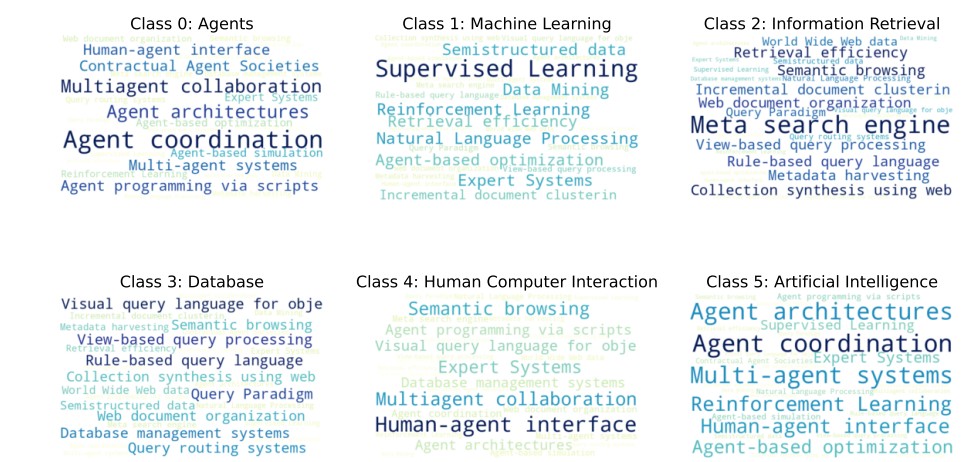

Figure 6: The average concept activations of 10 sampled instances per class across all selected concepts ($K = 30$) as word clouds on `Cora`.

Figure 7: The average concept activations of 10 sampled instances per class across all selected concepts ($K = 30$) as word clouds on `Citeseer`.

## E.2 Interpretability Study

We present word clouds in Figures 6, 7, 8, 9, and 10 to visualize the activation of all concepts from sampled instances across all classes and datasets. Additionally, we provide the complete versions of the Sankey diagrams for all datasets in Figures 11, 12, 13, 14, and 15.

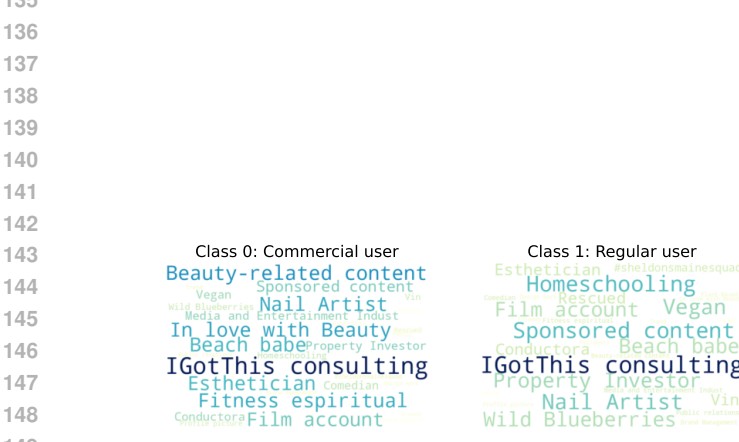

Figure 8: The average concept activations of 10 sampled instances per class across all selected concepts ($K = 30$) as word clouds on `Instagram`.

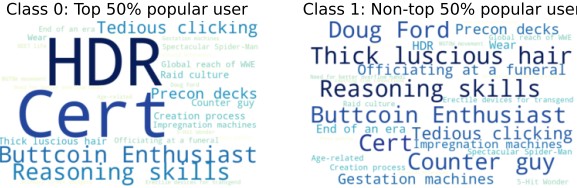

Figure 9: The average concept activations of 10 sampled instances per class across all selected concepts ($K = 30$) as word clouds on `Reddit`.

Class 0: Computational linguistics
Natural Language Processing
Computational Linguistics
Artificial intelligence
Interpreted Language
Text Mining
Language Modeling
Information retrieval
Scripting language
Algorithms Normalization

Class 1: Databases
Artificial Intelligence Algorithms
Directories
Natural Language Processing
Information retrieval
Normalization
Upper memory area NTFS
File allocation table
Text Mining Inodes
Memory management

Class 2: Operating systems
Memory management NTFS
Virtual memory BusyBox
User interface
Baikal CPU
Samsung Galaxy series
Live USB

Class 3: Computer architecture
Memory management
Upper memory area
Baikal CPU
Samsung Galaxy series
Virtual memory
Live USB Inodes
File allocation table

Class 4: Computer security
Security Expert
Inodes Secureworks
Password Cracking
Cryptography
Encryption
Artificial intelligence Malware
Electronic signatures

Class 5: Internet protocols
Electronic signatures
Normalization
File allocation table Baikal CPU
Encryption
Information retrieval
Secureworks NTFS
Artificial intelligence
Cryptography
Security Expert

Class 6: Computer file systems
File allocation table
Upper memory area
BusyBox Directories
Encryption Live USB
Inodes
NTFS
Baikal CPU Virtual memory

Class 7: Distributed computing architecture
Samsung Galaxy series
Algorithms Baikal CPU
Secureworks Security Expert
Web frameworks BusyBox
Information retrieval
Virtual memory Inodes
User interface Live USB
Memory management
Artificial intelligence

Class 8: Web technology
Natural Language Processing
Information retrieval
Scripting language
Text Mining Secureworks
Directories
Web frameworks
User interface Malware
Normalization

Class 9: Programming language topics
Artificial intelligence
Computational Linguistics
Interpreted Language
Language Modeling
Web frameworks User interface
Normalization
Scripting language
Algorithms Text Mining
Natural Language Processing

Figure 10: The average concept activations of 10 sampled instances per class across all selected concepts ($K = 30$) as word clouds on `WikiCS`.

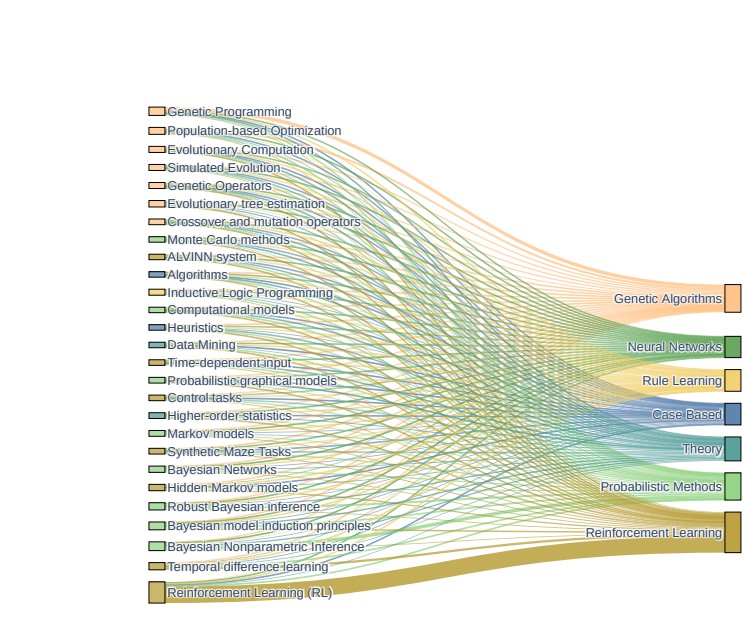

Figure 11: Sankey diagram for `Cora`

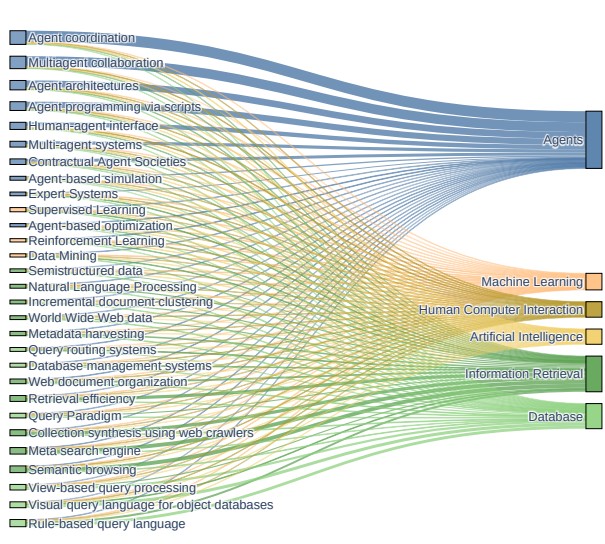

Figure 12: Sankey diagram for `Citeseer`

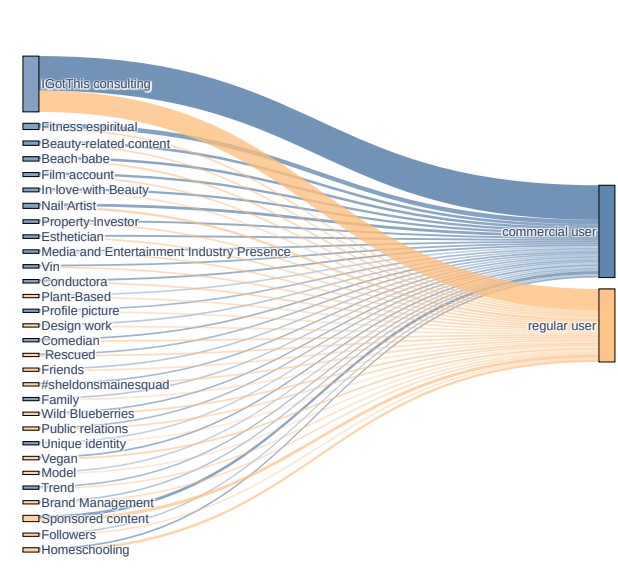

Figure 13: Sankey diagram for `Instagram`

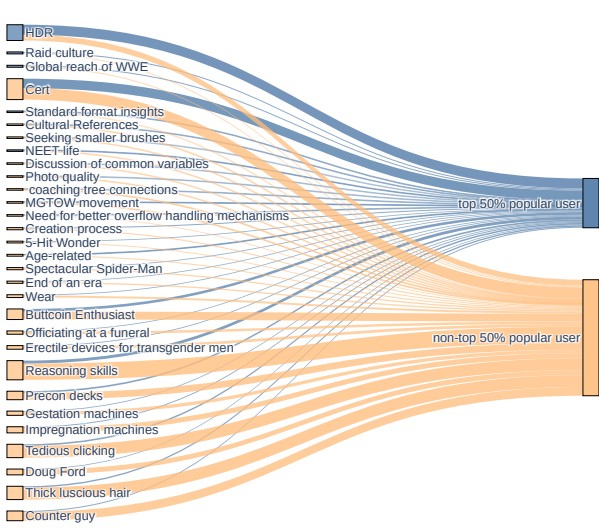

Figure 14: Sankey diagram for `Reddit`

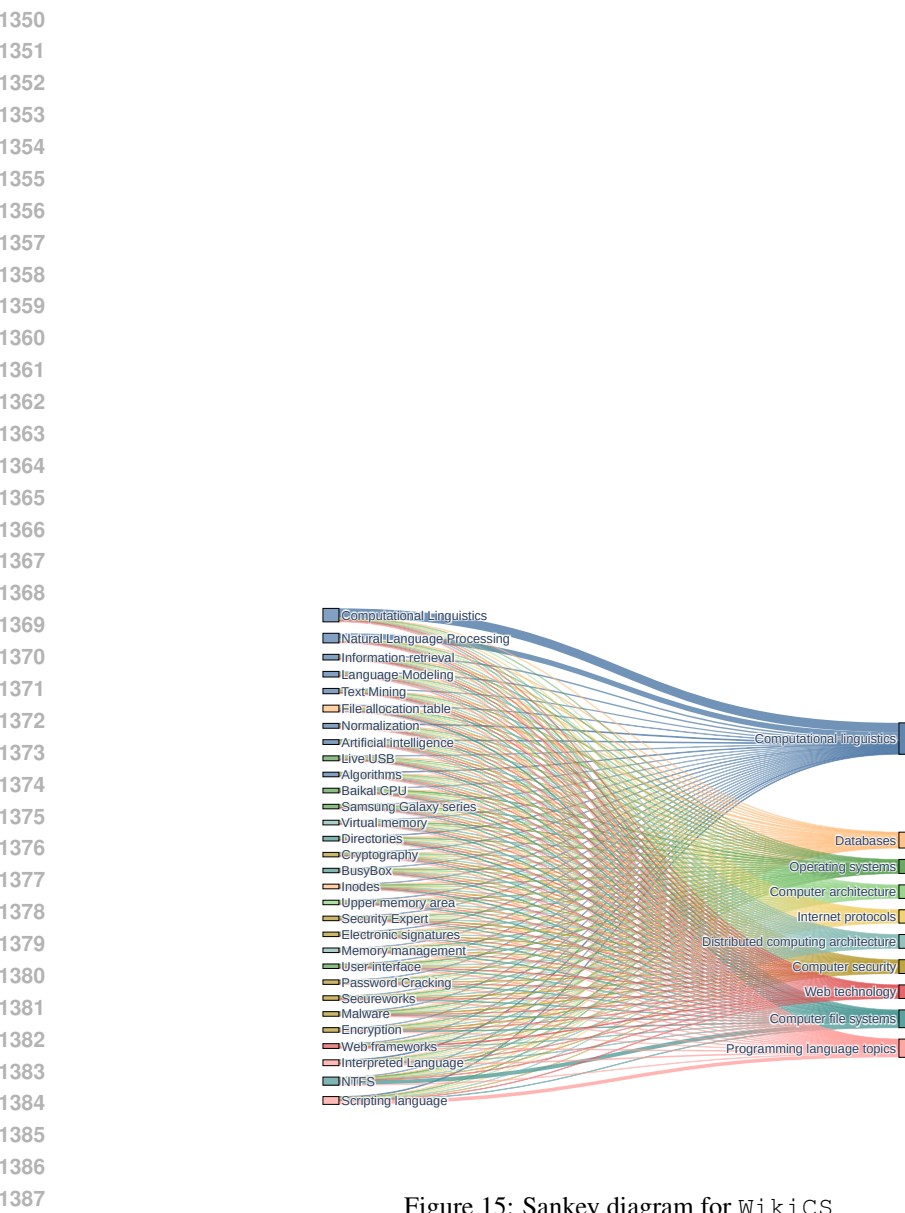

Figure 15: Sankey diagram for `WikiCS`

## F  THE USE OF LARGE LANGUAGE MODELS (LLMs)

In preparing this paper, we used large language models (LLMs) solely as a general-purpose tool to improve writing fluency and polish the presentation of the text. All ideas, experimental designs, analyses, and conclusions are our own, and the responsibility for the content rests entirely with the authors.

