# OpenReview forum: "Learning Language-grounded Concepts for Self-explainable Graph Neural Networks"
_ICLR.cc/2026/Conference — ICLR 2026 Conference Withdrawn Submission_

### Official Review · Reviewer_94io · 2025-10-17

**Soundness:** 1
**Presentation:** 1
**Contribution:** 2
**Rating:** 2
**Confidence:** 4

**Summary:**

The paper introduces a framework for building Graph Concept Bottleneck Models that leverage language-grounded concepts. The approach involves retrieving a set of concepts from graph nodes and using a contrastive loss to learn a shared embedding space between graph nodes and textual concepts. The goal is to align graph representations with semantic language concepts to enhance interpretability.

**Strengths:**

* The performances appear promising and competitive.

**Weaknesses:**

* **No evaluation on graph classification tasks**, limiting the assessment of the model’s generalization.
* The **method’s generality is overstated** — it is unclear how the proposed extraction mechanism would work across heterogeneous graph domains (e.g., molecules, social networks, citation graphs).
* **Concept extraction is poorly described** — the paper does not explain what information is provided to the LLM or how concepts are generated.
* **Lack of XAI-specific metrics** — there is no evaluation of fidelity, inverse fidelity, or sparsity; the interpretability claims are not validated (e.g., no analysis of important nodes or node–concept relationships).
* **Missing related work** — several key self-explainable GNNs and concept-based approaches (e.g., ProtGNN, PiGNN, Graph Concept Whitening, Global Concept Explanations for Graphs by Contrastive Learning) are not cited or discussed.
* The **strong performance might depend primarily on LLM embeddings**, rather than the proposed architecture itself.
* No visualization of concepts together with graphs.

**Questions:**

1. **Concept extraction:** How are the concepts obtained from the graph nodes, and what information is provided to the LLM during this process?

2. **Explainability evaluation:** Why are standard XAI metrics (e.g., Fidelity, InvFidelity, Sparsity) not included, and how do the authors validate that the learned concepts truly explain model behavior?

3. **Generality:** The approach is presented as general, but how can the same concept extraction mechanism adapt to graphs of very different nature (e.g., molecules vs. social networks)?

---

### Official Review · Reviewer_Qtvz · 2025-10-31

**Soundness:** 2
**Presentation:** 3
**Contribution:** 2
**Rating:** 4
**Confidence:** 3

**Summary:**

This paper introduces the Graph Concept Bottleneck (GCB), a novel framework for building self-explainable Graph Neural Networks (GNNs). To overcome the limited interpretability of traditional subgraph-based explanations, GCB forces the model to reason through an intermediate "concept bottleneck" layer composed of human-understandable natural language phrases. The framework consists of three main stages: 1) a universal graph encoder is pretrained using contrastive learning (CCGP) to align graph representations with a language-based concept space, 2) Large Language Models (LLMs) are used to automatically generate a rich, task-relevant set of candidate concepts, avoiding manual annotation, and 3) an Information Bottleneck-inspired objective is used to select a sparse, causal subset of these concepts for the final prediction. Because the model's output is a direct function of these concept activations, the resulting explanations are inherently faithful to the model's reasoning process. Experiments demonstrate that GCB achieves accuracy on par with black-box GNNs while offering significantly improved robustness to distribution shifts and data perturbations.

**Strengths:**

The paper presents a complete, end-to-end solution that cleverly addresses the key challenges of applying concept bottlenecks to graphs. The authors astutely identify that no "CLIP for graphs" exists and solve this by bootstrapping a graph-concept alignment model (CCGP) using LLM-generated annotations. The subsequent use of LLMs for automated, task-specific concept retrieval is a modern and practical approach that sidesteps the prohibitive labor cost of traditional concept-based models.

**Weaknesses:**

1. The core idea is a creative and powerful synthesis of existing work rather than a fundamentally new invention. Concept Bottleneck Models (CBMs) are an established paradigm, and the use of LLMs to generate concepts for CBMs has been explored in the vision domain. The novelty lies in the successful adaptation, integration, and engineering of these ideas to the graph domain, which presents unique challenges that the authors solve effectively.

2. Experiments:
- Potential for Information Leakage: The faithfulness analysis in Section 4.4 is good, but the results in Figure 3 raise some concerns. On the regular (in-distribution) splits, the performance gap between GCB and its random-concept variant (GCB-RC) shrinks as the number of concepts (K) increases. This suggests that with a large enough concept set, the model may begin to rely on spurious correlations in the patterns of concept activations rather than their actual semantics, which could undermine the faithfulness of the explanations.
- Limited Scope and Reliance on Text Attributes: All experiments are conducted on text-attributed graphs, where the connection between graph structure and language concepts is most direct. The paper speculates about applicability to other domains like molecular graphs, but the heavy reliance on LLMs for concept generation makes this extension non-trivial and unproven. The performance is fundamentally tied to the quality of the node text.
- Missing Ablations: The framework has several key components whose individual contributions are not fully teased apart. For example, concept retrieval uses both "Global" and "Instance-based" LLM prompting; an ablation showing the impact of each would clarify their necessity. Similarly, the CCGP pre-training is critical, but its quality is only evaluated implicitly through downstream performance.
3. Additional Things:
- High System Complexity: The overall GCB pipeline is quite complex, involving multiple stages: a separate pre-training phase, two distinct LLM-based concept retrieval steps, filtering, an Information Bottleneck optimization phase, and final predictor training. This represents a significant increase in complexity and computational cost compared to training a standard GNN.
- Dependency on External LLMs: The quality of the entire system is fundamentally dependent on the capabilities of the external LLM (GPT-3.5) used for generating both the pre-training annotations and the downstream concepts. This introduces a dependency on a large, proprietary black-box model and raises questions about reproducibility, cost, and potential biases inherited from the LLM.
- Missing literature: some related works are missing. Please add discussion on these papers[1][2][3].

[1] You W, Qu H, Gatti M, et al. Sum-of-parts: Self-attributing neural networks with end-to-end learning of feature groups[C]//Forty-second International Conference on Machine Learning. 2025.
[2] Wang S, Tang H, Wang M, et al. Gnothi Seauton: Empowering Faithful Self-Interpretability in Black-Box Transformers[C]//The Thirteenth International Conference on Learning Representations.
[3] Yu Y, Buchanan S, Pai D, et al. White-Box Transformers via Sparse Rate Reduction: Compression Is All There Is?[J]. Journal of Machine Learning Research, 2024, 25(300): 1-128.

**Questions:**

Please see weaknesses.

I would consider raising my scores if the asked experiments and discussion are involved properly.

---

### Official Review · Reviewer_K9EA · 2025-11-02

**Soundness:** 2
**Presentation:** 2
**Contribution:** 2
**Rating:** 4
**Confidence:** 3

**Summary:**

The paper introduces a framework (GCB) that enhances interpretability in graph neural networks by bridging graph learning and natural language understanding. In this process, GCB embeds graphs into a language-based concept bottleneck, where each concept corresponds to a natural-language phrase. The framework integrates three modules: A. Contrastive Concept–Graph Pretraining (CCGP) for aligning graph and text representations, B. LLM-based Concept Retrieval for automatically generating candidate concepts without manual labels, and C. Information-Constrained Bottleneck Optimization to retain causal concepts while filtering spurious ones. Authors conducted experiments that demonstrate GCB achieves comparable accuracy to black-box GNNs, while improving robustness and human interpretability under distribution shifts and data perturbations.

**Strengths:**

1. The paper is well motivated, it tries to make GNNs self-explainable through language-grounded concepts. The architecture is reasonably organized, and each module is explained with clarity.

2. While building on existing paradigms such as concept bottleneck models and contrastive pretraining, GCB extends them into the graph domain and grounds concepts in natural language, which is an interesting aspect of graph interpretability.

3. The paper shows consistent results across multiple benchmarks, and the experimental study provides a convincing level of completeness. Besides, the work appears practically valuable by demonstrating that interpretability can be achieved without sacrificing predictive performance, makes it as a feasible framework that could be applied to real-world graph learning tasks.

**Weaknesses:**

1. Although GCB connects graphs with language concepts, it seems not able to show which specific nodes or subgraphs support a prediction. Without linking concepts back to the graph structure, the explanations remain abstract and hard to verify.

2. The author's approach depends on GPT-generated concepts, which might include noisy or redundant semantic descriptions. It would be helpful if the paper analyzed how sensitive the results are to prompt design or explored whether these concepts truly reflect underlying graph patterns rather than surface correlations.

3. The paper did not examine if GCB maintains interpretability on larger, more complex graphs where concept meanings may overlap or lose precision.

**Questions:**

1. Unlike prior GNN explanation methods (e.g., GNNExplainer [1], PGExplainer) that visualize subgraph-level importance, this paper does not provide any instance-level structural visualization. Since GCB focuses on language-grounded semantic concepts, could the authors clarify whether structural visualization is possible or how semantic concepts correspond to subgraph patterns in practice?

[1] Ying, Zhitao, et al. "Gnnexplainer: Generating explanations for graph neural networks." Advances in neural information processing systems 32 (2019).

[2] Luo, Dongsheng, et al. "Parameterized explainer for graph neural network." Advances in neural information processing systems 33 (2020): 19620-19631.

2. How do the authors ensure that the grounded language concepts correspond to causal or structurally relevant features in the graph, rather than abstract textual correlations introduced by the LLM (any verifications)?

3. Does GCB maintain interpretability when applied to large, real-world graphs, where concept meanings may overlap or become unclear? E.g., in molecular graphs, concepts like “contains ring” and “has cyclic structure” might refer to almost the same pattern, and in social graphs, “dense community” and “close friendship group” could overlap. How does GCB handle such ambiguity between similar concepts?

---

### Note · Authors · 2026-01-22

I have read and agree with the venue's withdrawal policy on behalf of myself and my co-authors.